# Towards a potent and rapidly reversible Dexmedetomidine-based general anesthetic

**Zheng Xie**[1]*, **Robert Fong**[1], **Aaron P. Fox**[2]

**1** Department of Anesthesia and Critical Care, The University of Chicago, Chicago, IL, United States of America, **2** Department of Neurobiology, Pharmacology and Physiology, The University of Chicago, Chicago, IL, United States of America

* jxie@bsd.uchicago.edu

## Abstract

Clinically useful anesthetics are associated with delirium and cognitive decline in the elderly. Dexmedetomidine (Dex), an $\alpha_2$ adrenergic receptor agonist, is an intravenous sedative with analgesic properties. Dex is associated with a lower incidence of delirium in the elderly. In this study, we first assessed whether a high dose of Dex alone was a clinically useful anesthetic. Finding that it was not, we sought to determine whether supplementation of Dex with low doses of two common anesthetics, propofol or sevoflurane, created an effective general anesthetic. Rats were sedated with a bolus followed by a continuous infusion of Dex and a low dose of a second agent—propofol, or sevoflurane. A strong noxious stimulus was applied every 15 minutes while monitoring vital signs. A combination of the $\alpha_2$ competitive antagonist, atipamezole, and caffeine was administered to reverse the anesthesia. Abdominal surgery was used to validate the efficacy of these dosing regimens. The animals responded to noxious stimuli when receiving Dex alone. Supplementing Dex with either a low dose of propofol or sevoflurane completely suppressed responses to the noxious stimulus and allowed the rats to tolerate abdominal surgery with complete immobility and no alterations in vital signs, suggesting that the drug combinations were effective anesthetics. EEG recordings showed suppression of high frequency activity suggesting that awareness and memory were impaired. Previously we found that combination of atipamezole and caffeine rapidly and completely reversed the sedation and bradycardia elicited by Dex. In this study, atipamezole and caffeine accelerated the time to emergence from unconsciousness by >95% in Dex supplemented with either propofol or sevoflurane.

## In conclusion

Our results suggest that Dex supplemented with a low dose of a second agent creates a potent anesthetic that is rapidly reversed by atipamezole and caffeine.

**Data Availability Statement:** All relevant data are within the paper and its Supporting Information files.

**Funding:** This study is supported by a NIH grant (GM-116119) To ZX and APF. The funders had no

role in study design, data collection and analysis, decision to publish, or preparation of the manuscript.

**Competing interests:** The authors have declared that no competing interests exist.

**Abbreviations:** Dex, dexmedetomidine; Ati, atipamezole.

## Introduction

Anesthesia in current clinical practice is exceptionally safe [1]. As such, anesthesia research has focused on optimizing currently available anesthetics rather than developing new agents [1]. Many of the anesthetics currently in use were developed decades ago [2–8]. Despite their favorable safety profile, these anesthetics, including sevoflurane and propofol, are associated with delirium and cognitive dysfunction in the elderly [9–13]. Additionally, neuroapoptosis and cognitive alterations were observed in animal studies that included non-human primates [14–20]. Although unambiguous evidence of neurotoxicity in humans remains to be demonstrated, the search for new and potentially safer anesthetic regimens represents a valuable endeavor. Dexmedetomidine (Dex) is a sedative associated with a lower incidence of delirium and cognitive complications in the elderly [10, 21–23] and neural protection in the young [24, 25]. Our goal in this study was to determine whether Dex alone, at high dosages, could produce an effective anesthetic. If Dex alone was not an effective anesthetic, then we sought to determine whether supplementing Dex with low, subanesthetic doses of two common anesthetics, propofol or sevoflurane, produced an efficacious intraoperative general anesthetic.

A successful anesthetic embodies four cardinal traits; amnesia, unconsciousness, antinociception and immobility [26]. Amnesia is difficult to study in animal models, while unconsciousness, antinociception and immobility are readily quantified. While Dex is an effective sedative, it is a poor amnestic and immobilizer at the concentrations used clinically.

First, high dose Dex was tested to determine whether rats remained unconscious, unresponsive and immobile when exposed to a powerful noxious stimulus. Next, low doses of two common anesthetics, propofol and sevoflurane, were used to supplement Dex. Although propofol and sevoflurane have been implicated in neurotoxicity, high doses of the anesthetics were required to initiate apoptosis and cognitive decline in young animals. The low doses of these anesthetics employed in this study fall below the range shown to engender these deleterious effects. Furthermore, Dex seems to mitigate apoptosis caused by other anesthetics [27, 28], suggesting that Dex may be neuroprotective [29–31]. Future neuroapoptosis and behavioral studies will be required to confirm the safety of the drug combinations used in our study.

Both drug combinations, Dex/propofol or Dex/sevoflurane, suppressed responses to a powerful noxious stimulus, while high dose Dex alone did not. Furthermore, Dex with sevoflurane or propofol suppressed all motor and autonomic responses during abdominal surgery. Propofol, and sevoflurane are amnestic at low concentrations [32–34]: Our EEG recordings suggest that memory is impaired by the anesthetic combinations we tested. Dex with low-dose sevoflurane or propofol produced less burst suppression than sevoflurane alone near its $EC_{50}$.

We have shown previously that a combination of low dose atipamezole and caffeine reverses Dex sedation with remarkable effectiveness [35]. This reversal cocktail accelerated emergence from unconsciousness produced by Dex supplemented with propofol or sevoflurane with equal efficacy. Our results suggest that these drug combinations based primarily on Dex meet the requirements for an effective general anesthetic that is rapidly reversed by low dose atipamezole and caffeine.

## Materials and methods

### Ethics and animals

This animal study was approved by The University of Chicago Institutional Animal Care and Use Committees (protocol #42437). This manuscript adheres to the applicable ARRIVE guidelines. Between experiments animals were cared for by University of Chicago veterinary staff. Forty female Adult Sprague Dawley rats (Charles River, Wilmington, MA), weighing 250–400

gm and 8 male rats weighing 250–350 grams were used in the study. They were transported to the anesthesia room for multiple anesthesia sessions with at least five days in between sessions. At the completion of each anesthesia session, rats were transported back to their own home room. Rats were divided into groups of 8 for each set of experiments. Each rat was never sedated more than 6 times. All rats served as their own controls. All experiments were performed during the daytime and at room temperature of 22–27˚C. While on a nose cone and throughout the study rats were placed on a heating pad at 25˚C. During experiments, heart rate, respiratory rate and blood oxygen saturation were monitored with a Kent Scientific PhysioSuite. $SpO_2$ was always >92% throughout the experiments. In 2 subsets of experiments, blood pressures (BP) from the tails of the rats were measured by a BP system (IITC Life Science Inc., CA). Noxious stimuli were stopped at the first sign of distress. At the conclusions of the study, rats were euthanized by the animal facility staffs using $CO_2$ overdose, followed by decapitation. In experiments where surgery was performed, the rats were sacrificed, after the wound were closed with sutures, by an overdose of propofol (20 mg/kg) and decapitation by veterinary staff.

## Calibrated noxious stimulus

A calibrated tail clamp was used to assess anesthetic efficacy. The stimulus was generated with Kelly forceps that were used to clamp each rat's tail where it was exactly 5 mm in diameter. Clamping the tail to the first stop on the forceps for 30 seconds, or until a response was evoked, provided a consistently reproducible stimulus. Elevated levels of sevoflurane was required to suppress motor response to the tail clamp.

## Determination of minimum alveolar concentration (MAC) equivalence of sevoflurane in suppressing tail clamp stimulus

Minimum alveolar concentration (MAC) is used to quantify the potency of inhaled anesthetics, representing the $EC_{50}$ for motor response to a noxious stimulus [36, 37]. The rats were initially anesthetized by placement into a gas-tight chamber (volume 6 liters) into which sevoflurane was delivered by an anesthesia machine (Ohmeda Modulus II Plus). A nose cone was connected to the outlet port of the gas chamber by a short length of corrugated tubing. The gas concentration was sampled at the outlet port of the gas chamber by a gas analyzer (Intellivue MP70, Philips). After each rat was anesthetized in the gas chamber with 3.3% sevoflurane with $2LO_2/2LAir$ for 10 minutes, the rat was weighed and then placed with its face in the nosecone. A 24g intravenous catheter was placed in a tail vein and the rat was anesthetized with 3% sevoflurane with $1LO_2/1LAir$ via the nose cone for another 20 minutes to reach steady state. The initial anesthetic concentration was chosen as 1.2–1.3 MAC as determined by other studies [38–40]. The tail clamp was applied as described above. Depending on the rat's response to the tail clamp, sevoflurane was dialed up for positive response or down for negative response by 0.3%. The rat was anesthetized with the next concentration of sevoflurane for 15 minutes before repeating the tail clamp. The test would end if the rat stopped responding to the tail clamp on the step up or responded to the tail clamp on the step down. The concentration where 50% of rats did not respond to the tail clamp was considered the MAC for sevoflurane.

## Drugs

Caffeine (Sigma-Aldrich, St Louis, part # C0750-5G, Lot#SLBD0505V) was dissolved in sterile saline to a final concentration of 10 mg/ml, and rats were dosed intravenously to a final dose of 25mg/kg. Sterile saline injection was used as vehicle control for caffeine.

Atipamezole (also called Antisedan) was manufactured by Zoetis Pharmaceuticals, Parsippany, NJ (#RXANTISEDAN-10). The same bottle was used for the entire study. The atipamezole dosages used in the studies outlined in the manuscript varied from 5 µg/kg to 20 µg/kg. Atipamezole was administered intravenously to the rats with sterile saline as the vehicle at 5 µg/ml.

Dexmedetomidine (Dexmedetomidine hydrochloride: NDC 16729-239-93) was purchased from Accord Healthcare, Durham, NC, at a 5 µg/ml concentration in saline. Dex was delivered intravenously by an infusion pump (Medfusion 4000, Smith Medical ASD, Inc. St. Paul, MN).

Propofol (Diprivan injectable emulsion NDC 63323-269-29) was purchased from Fresenius (Kabi, Lake Zurich, IL) at a 10 mg/ml concentration. The drug was delivered by an infusion pump.

Sevoflurane (Ultane, NDC 0074-4456-04) was purchased from Abbvie, North Chicago, IL.

## Sedation/ Anesthesia

Rats were placed in a gas-tight anesthesia chamber where they were exposed to either 1.8% isoflurane or 3.3% sevoflurane (in 2L/min $O_2$ & 2L/min Air) for 10 minutes, rendering them unconscious and insensitive to mild tail pinch. Rats were then removed from the gas tight chamber and weighed. Anesthesia was maintained with 1.8% isoflurane or 3.3% sevoflurane in 1L/ min $O_2$ & 1 L/min Air, delivered via a nose cone. A 24g intravenous (IV) catheter was inserted into a tail vein. Isoflurane was used to render rats' unconscious such that they could be weighed and an I-V line inserted. In some experiments, EEG electrodes were inserted, and the EEG recorded under isoflurane was used as a baseline. Isoflurane was terminated after the bolus dose of Dex or Dex with propofol was delivered. The tail clamp stimulus was applied only after the washout of isoflurane was complete. In sevoflurane experiments we used sevoflurane throughout. The IV was inserted after 20 minutes 3.3% sevoflurane, followed by 1.4% sevoflurane with Dex bolus and infusion.

## Dex infusion with and without a second agent

Dex was administered via an infusion pump attached to the IV line. A bolus of Dex was delivered over 5 minutes via a pump. This was followed by 60 minutes of continuous infusion of Dex (see Fig 1). Every 15 minutes following the bolus, vital signs were recorded, and the response to tail clamp assessed. The rats breathed 1L/min $O_2$ & 1 L/min Air for the entire duration of Dex exposure.

The second lumen of a Y shaped microcatheter (Baxter Healthcare Corp, Deerfield, IL) allowed us to use a second infusion pump for propofol. Reversal agents or saline were injected by syringe connected directly to the IV catheter and flushed with 0.5 ml saline. The IV catheter was then removed, and the rats were placed in a cage on their backs. The time to emergence was defined as the time required for the rats to right and stand on all 4 paws. (Also referred to as recovery of righting reflex–RORR).

## Electroencephalogram (EEG) recording

Scalp electrodes were used for electroencephalographic recording. To avoid stress effects following invasive surgical implantation of electrodes, we employed 9 mm stainless steel EEG needle electrodes inserted into the scalp during anesthesia so that they were touching the outer table of the skull. After rats were anesthetized with 3.3% sevoflurane or 1.8% isoflurane, two scalp electrodes [Astro-Med/ Grass Technologies] were placed, as shown in S1 Fig. We drew a line between the anterior edge of bilateral ears, between Bregma and Lambda as described in reference [41]. From the midpoint, one electrode was placed anteriorly perpendicular to the

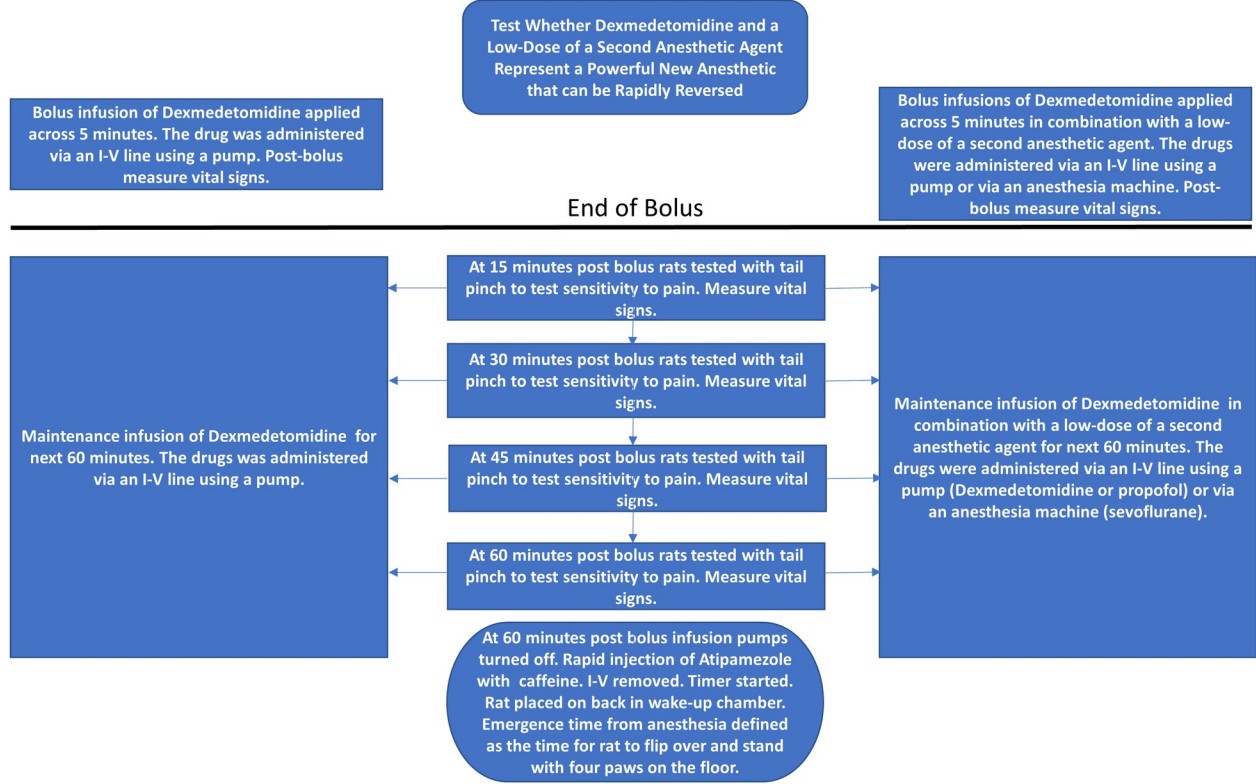

**Fig 1. Experimental protocol.** For this protocol, a 5-minute infusion of a bolus of Dex (10 μg/kg) was followed by 60 minutes of a maintenance infusion of Dex (10 μg/kg/hr), left panel. The same rats were exposed to a similar protocol, but one where a second agent supplemented the Dex, either propofol or sevoflurane, right panel. The second agent was present for the entire Dex infusion. At various times, the rats were tested with a tail clamp to measure immobility. Vital signs were obtained throughout the experiment. At the end of the protocol, rats received an injection of either saline or atipamezole with caffeine (randomized order). Rats were then placed on their backs in a waking box, and the time for the rats to recover their righting reflex was recorded. This time is plotted in the subsequent Figs as the emergence time.

line and the other posteriorly perpendicular to it. Two EEG channels were recorded, the first one (red) from an electrode placed over the anterior portion of the brain, and a second electrode (green) placed over the posterior portion of the brain. The EMG lead (yellow) was obtained from an electrode placed over the left shoulder, all referenced to an electrode (white) placed near medial to the ears. A ground electrode (black) was placed on the opposite side of the reference lead. These three channels were recorded with an A/D rate of 500 Hz/channel, with a 0.05–100 Hz bandpass and 12 dB/octave roll-off. Potentials were amplified with a Neuroscan SYNAMPS 2 system (Compumedics, Inc., Charlotte, NC).

## Power spectra

Two types of power spectra were computed: conventional power spectra using the SYNAMP EDIT module, and spectrograms using MATLAB R2021, and the EEGLAB program, using time resolution of 10 seconds with 99% overlap. Power spectra (dB) were computed over 5-minute-long epoch of EEG, partitioned into 512-point epochs, and averaged, yielding a temporal resolution of 2 Hz. Power was calculated as the fraction of a specific frequency power, including delta (0.5–4 Hz), theta (4–8 Hz), alpha (8–12 Hz), spindle (12–15 Hz) and beta (15–25 Hz) by MATLAB R2021. The average powers of 8 rats in each frequency band were compared in the different period of times between sevoflurane 3.0% sevoflurane and 1.4% sevoflurane with Dex infusion in one group and between 1.7% isoflurane and propofol and Dex

infusion in another group. Burst suppression ratio (BSR) in the EEG was calculated by a formula, BSR = (total time of suppression/epoch length) x 100% and analyzed independently by two different members of this study. Each independent analysis produced consistent results. Suppression time was defined from 0.5 to 5 seconds consistent with other studies [42–44]. Burst suppression in EEG was defined as an amplitude < 5 µV which lasted for ≥ 30% of each minute.

Power spectra were obtained for two 5-minute periods and reference two separate conditions during the session. In one group, animals were exposed to sevoflurane 3.0%, then sevoflurane was turned down to 1.4%, then Dex bolus10 µg/kg and infusion 12 µg/kg/hr started. Power spectra were obtained under each condition, including after forty minutes of the subanesthetic dose of sevoflurane together with Dex infusion. In another group, the animals were first exposed to isoflurane 1.7% for 25 minutes, then isoflurane was turned off and washed out, and propofol 4 mg/kg plus Dex bolus10 µg/kg were given over 5 minutes and followed by propofol 300 µg/kg/min and Dex 12 µg/kg/hr infusion. Power spectra were obtained under each condition including forty minutes into subanesthetic dose of propofol with Dex infusion. During these anesthesia sessions, the global changes in EEGs were more prominent anteriorly. Therefore, we used the signal recorded from the anterior lead for analysis in this study.

## Statistical analysis

The sample size used in this study was based on an analysis described in a previous study from our lab and by using GPower [35]. In this study the threshold for statistical significance was set to 0.05. The statistical test used to analyze each data set is described in the appropriate figure legends. If three or more comparisons were required within a group of animals a repeated measures analysis of variance (RM-ANOVA) with Tukey's multiple comparisons post-hoc test was employed. Data was evaluated for normality. When only 2 conditions were assessed, either a paired or an unpaired T-test was employed Data was analyzed and graphed using GraphPad Prism 9 software. Data were expressed and plotted graphically as mean ± standard deviation (SD).

The experiments shown in this manuscript were done in an unblinded manner. Experimental order and drug application were randomized.

All data obtained in this study were shown in this manuscript and supporting information.

## Results

### Is high dose of Dex an effective anesthetic?

To assess whether high doses of Dex suppressed responses to a tail clamp, we subjected 8 rats on separate days to either a high dose or low dose infusion regimen. In the high dose regimen, we administered a 40 µg/ kg bolus of Dex over 5 minutes followed by a continuous infusion of Dex at 48 µg/ kg/ hr for an additional 60 minutes. In the low dose regimen, we administered a 10 µg/ kg bolus over 5 mins followed by a continuous infusion of Dex at 12 µg/kg/hr for an additional 60 minutes.

In animal studies of anesthetic efficacy, ablation of motor response to a noxious stimulus is typically assessed. Such a stimulus was generated by tail clamping as described in the Methods section. Every 15 minutes a painful tail clamp was applied to the rats, while monitoring the animals for any responses including vital sign changes. S1 Table compares the responsiveness to the tail clamp at various timepoints with these two dosing regimens in female rats. Aggregating the data from all 4 time points, S1 Table shows that there were only two positive responses to the tail clamp out of 32 total tests when animals received high dose Dex, compared to 25

positive responses out of 32 total tests when those same animals received low dose Dex ($p < 0.0001$, Fisher's exact test).

In a previous study we observed that after a single bolus of Dex at 40 μg/kg, rats required approximately 100 mins to regain their righting reflex as compared to 30 mins after a 10 μg/kg Dex bolus [35]. The infusion used in the current study should produce similar extended emergence times. Fig 2 shows that atipamezole and caffeine reversed the unconsciousness engendered by high dose Dex. While the rats required $10,891 \pm 5928$ seconds to emerge from unconsciousness following a control saline injection administered after terminating the high dose Dex infusion, they required only $17 \pm 11.24$ seconds to emerge after injection of 20 μg/ kg of atipamezole and 25 mg/ kg caffeine ($p = 0.013$, paired t-test). Although the rats recovered their righting reflex, they remained sluggish for an extended period. Thus, reversal was incomplete.

High dose Dex infusion was associated with a decrease in heart rate and respiratory rate, while blood oxygen saturation remained unchanged (Fig 3). Using a 2-way repeated measures ANOVA we found no difference in vital signs when comparing high dose to low dose Dex regimens in the same animals at the same time points. This result suggests that the cardiovascular changes observed following the administration of Dex saturate at the lower dosage.

After administration of the Dex boluses, HR was significantly lower for all subsequent times compared to that observed prior to application of Dex. The RR was significantly depressed by Dex, but just for a single time point. The RR recovered by 30 minutes after the Dex bolus and remained back at baseline levels for subsequent measurements. There was no change in $SPO_2$ elicited by the application of Dex.

## Dex and low dose propofol

The goal of these studies was to determine whether a low dose of a second agent in combination with Dex produced an effective anesthetic. We define a "low dose" of the second agent as one incapable of maintaining unconsciousness as assessed by the ablation of the righting reflex. Rats received an intravenous bolus of 5 mg/kg of propofol over 5 minutes followed by a continuous infusion at varying concentrations for an additional 60 minutes. S2 Fig shows the time to emergence at various infusion doses of propofol. As is evident in S2 Fig, a minimum infusion dosing of 400 μg/kg/min was needed to maintain unconsciousness. At 300 μg/kg/min, all animals spontaneously emerged from anesthesia prior to the cessation of the hour-long infusion. Because these propofol levels were sub hypnotic, 300 g/kg/min or less, they were considered "low dose".

S2 Table shows the results of an experiment in which a low dose of propofol was used to supplement Dex. Rats were first anesthetized with isoflurane (1.8%) to allow intravenous cannulation. They then received a bolus of both Dex (10 μg/ kg) and of propofol (4 mg/ kg) over 5 minutes, after which the isoflurane was terminated, and the rats breathed an $O_2$/Air mixture. Over the next 60 minutes, the animals received infusions of Dex (15 μg/ kg/ hr) and propofol (200 μg/ kg/ min). We applied a tail clamp every 15 minutes to the rats while recording vital signs. At the end of the hour the infusions were discontinued, and the animals were injected with either saline (control) or with atipamezole (20 μg/ kg) and caffeine (25 mg/ kg). We then placed the rats on their backs to assess the time for recovering their righting reflex.

S2 Table shows that aggregating the data from all 4 time points, that for Dex alone the rats responded to the tail clamp 24 out of 32 times tested, compared to only 3 out of 64 times for Dex and low dose propofol ($p < 0.0001$, Fisher's Exact Test). This dose of propofol when combined with Dex was insufficient to completely suppress responses to the tail clamp.

We initially tested Dex at a dose we employed in our previous study [35] with propofol at 200 μg/kg/min. This dose was not able to ablate tail clamp response in all animals tested.

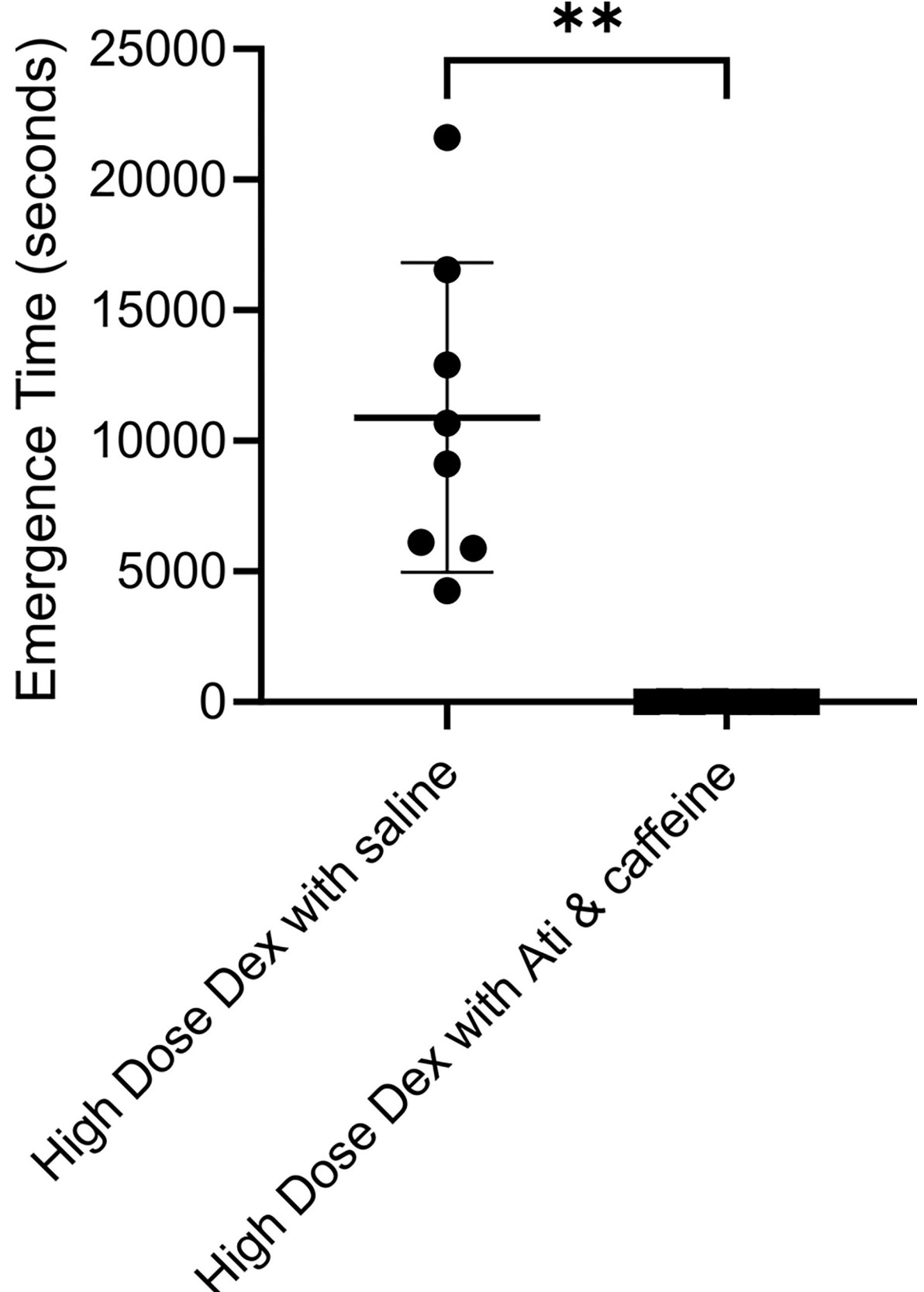

**Fig 2. The combination of atipamezole and caffeine dramatically accelerated emergence from the anesthesia produced by a high dose of Dex used by itself.** The same group of 8 rats were exposed to two Dex's sessions, a week apart. At the end of one session the rats received a bolus injection of saline and in the other atipamezole (20 μg/kg) and caffeine (25 mg/kg). The order of the drug injections was randomized. Rats were placed on their backs in a waking box, and the time for the rats to recover their righting reflex was recorded. Saline injected rats are plotted on the left while atipamezole and caffeine injected rats are plotted on the right. The >99% difference in emergence time was significant (p = 0.013, two tailed paired T-test, t = 5.196 and df = 7).

Therefore, we repeated the experiment with a higher dose of propofol and slightly lower dose of Dex.

When the experiment was repeated with an infusion of 300 μg/kg/min of propofol to supplement the Dex at 12 μg/kg/hr, none of the rats responded to the tail clamp at any time point (0 responses out of 32 tests, S3A Table). This difference was significantly different than Dex by itself (p < 0.0001, n = 32). From the data shown above, 300 μg/kg/min represents the lowest possible dose of propofol to supplement Dex if the goal is to achieve complete suppression of any response to the tail clamp. These results suggest that a Dex infusion of 12 μg/kg/hr with a second agent is sufficient to produce an effective immobilizing and antinociceptive agent. Therefore, Dex infusion at 12 μg/kg/hr was used as the base in subsequent Dex combinations in the present studies.

Repeating the same experiment with a cohort of 8 male rats showed an identical result. When Dex was used by itself, the rats responded to the tail clamp 31 times out of 32 total tests (S3B Table). When Dex was supplemented with the low dose of propofol (4 mg/ kg bolus, then infusion of 300 μg/kg/min) none of the rats responded to the tail clamp at any time point. Reponses to the tail clamp in Dex alone were significantly different than those in Dex supplemented with low dose propofol (p < 0.0001, Fisher's exact test).

Prolonged emergence time has been a significant drawback to the use of Dex infusions in clinical practice. Previously we showed that atipamezole and caffeine effectively reversed sedation engendered by Dex. In this experiment we tested whether atipamezole and caffeine could reverse the combination of Dex and low dose propofol. Fig 4 shows that it took the female rats 3284 ± 786 (mean ± SD) seconds to emerge from unconsciousness when they received a control saline injection, but a mere 141.9 ± 123.2 seconds to emerge after injection of atipamezole (20 μg/ kg) and caffeine (25 mg/ kg) a >95% difference which was significant (p < 0.0001, two tailed paired T-test, t = 11.89 and df = 7).

Fig 3 shows that atipamezole (10 μg/kg) & caffeine (25 mg/kg) were equally effective at accelerating emergence in male rats exposed to Dex & propofol (4 mg bolus/ 300 μg/kg/min infusion). Emergence was rapid, occurring at 105.4 ± 75.4 sec (mean ± SD). We previously showed that male rats woke in ~2700 seconds for this dose of Dex when used by itself [35].

The combination of Dex and propofol used to generate the data in S3 Table and Fig 4 was associated with a statistically significant decrease in heart rate and respiratory rate, while blood oxygen saturation remained unchanged (Fig 5). Vital signs recorded with Dex alone were not different from those recorded in Dex with propofol.

## Determination of EC50 or minimum alveolar concentration (MAC) equivalence of sevoflurane

The minimum alveolar concentration (MAC) is defined as the concentration of an inhalational anesthetic agent that suppresses response to a noxious stimulus in half of a test population [36, 45]. In published studies, 2.0% - 2.4% sevoflurane was identified as ~1 MAC in adult rats [38, 39]. S4 Table show that in our study, 3.0% sevoflurane was required to suppress response to tail clamping in 50% of rats tested and thus represents 1 MAC. The most likely explanation for the difference in MAC between our results and that in the literature is that we

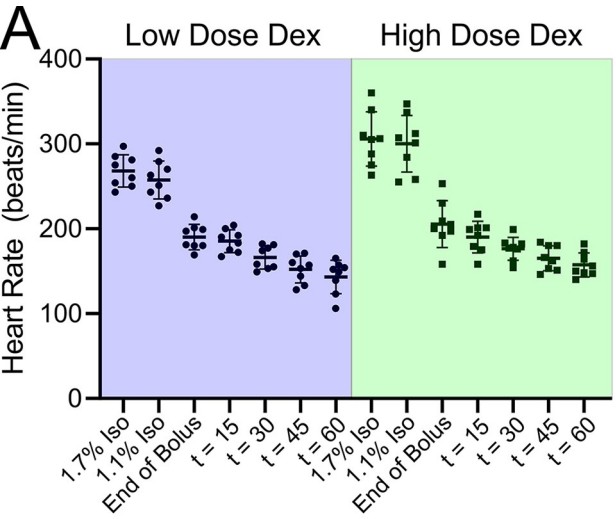

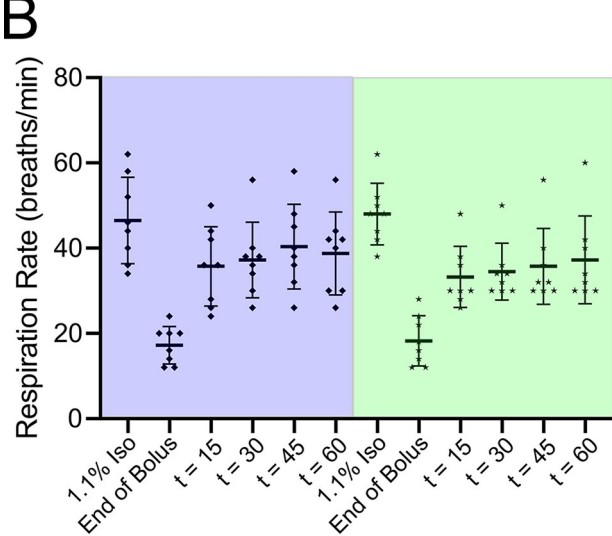

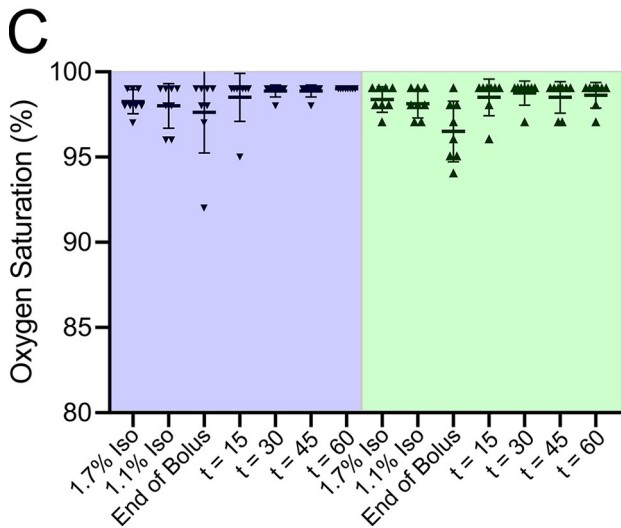

**Fig 3. Rats exposed to high-dose Dex by itself exhibited slowed heart and respiration rates while leaving blood oxygen saturation unaffected.** The vital signs were measured at various times during the experiment. The first measurement ("1.7% isoflurane") took place while the rats were receiving 1.7% isoflurane, prior to Dex administration. The second measurement ("1.1% isoflurane") was taken while the rats were receiving 1.1% isoflurane, prior to Dex administration. The third measurement ("End of Bolus") was taken immediately after the bolus of Dex was administered. The 1.1% isoflurane was turned off but not yet washed out. Rats were breathing $O_2$. The fourth measurement ("t = 15") was taken fifteen minutes after the end of the bolus. All isoflurane should have been washed out since rats had been breathing $O_2$ for 15 minutes. The fifth, sixth and seventh measurements ("t = 30", "t = 45", "t = 60") were taken thirty, forty-five and sixty minutes after the end of the bolus. The green panels represent high dose Dex (bolus 40 μg/kg: infusion of 48 μg/kg/hr) while the blue panel data was obtained from the same rats exposed to a lower dose of Dex (bolus 10 μg/kg: infusion 12 μg/ kg/ hr). Comparisons of HR at different times for Dex 10/12. Dex 40/48 was similar. For this analysis a repeated measures two-way ANOVA was employed: 1.7% isoflurane vs. 1.1% isoflurane, p = ns: We compared the HR at 1.1% isoflurane to the rest of the time points, 1.1% isoflurane vs. End of Bolus, p = 0.002: 1.1% isoflurane vs. t = 15, p = 0.002: 1.1% isoflurane vs. t = 30, p <0.0001: 1.1% isoflurane vs. t = 45, p < 0.0001: 1.1% isoflurane vs. t = 60, p <0.0001. Comparisons of RR: Only the following times were different. 1.1% isoflurane vs. End of Bolus, p = 0.0003: End of bolus vs t = 15, p = 0.01: End of bolus vs t = 30, p = 0.01: End of bolus vs t = 45, p = 0.004: End of bolus vs t = 60, p = 0.02. Comparisons of $SpO_2$: No significant changes were observed. We found no evidence for a time by condition interaction (Dex 10/12 and Dex 40/48). Subsequent low Dex versus high Dex at each time point comparisons gave the following: no significant difference in HR for any time points: 1.7% isoflurane, 1.1% isoflurane, End of bolus, t = 15, t = 30, t = 45 and t = 60 between the two conditions with adjusted p = 0.64, p = 0.71, p = 0.99, p = 0.99, p = 0.99, p = 0.99 and p = 0.99, respectively. There was also no evidence for a time by condition interactions between the two conditions in RR and SpO2 at any time points.

measured gas concentrations in the corrugated tubing leading to the rat's nose cone and not alveolar levels. Previous studies have shown that measurement differences give rise to significantly different MAC values. For example, MAC values using anesthesia box gas values were different than those measured assaying alveolar gas concentrations at the tracheotomy site [46].

## Dex and low dose sevoflurane

Dex was next paired with 1.4% of sevoflurane (< 0.5 MAC), considered to be "low dose" based on our experimental determination of a MAC of 3% as outlined above. At this dose of sevoflurane alone, most rats would move or right themselves with a mild stimulus such as IV insertion. Further reducing sevoflurane dose might lead to the recovery of righting reflex (RORR) in some rats before the completion of Dex bolus. As before, a bolus of Dex (10 μg/kg) given over 5 minutes, followed by a continuous infusion of Dex (12 μg/kg/hour) for the next 60 minutes.

S3 Table shows that aggregating data in female rats from all 4 time points, no animal responded to the tail clamp in when Dex was paired with low-dose sevoflurane, 32 total tests, while in rats treated with Dex alone, 25 responded in 32 total tests (P<0.0001).

In female rats, Atipamezole (10 μg/ kg) and caffeine (25 mg/ kg) rapidly reversed Dex alone (9.5 ± 10.99 sec, n = 8) as well as the combination of Dex and sevoflurane (146.3 ± 46.60 sec, n = 8), although it was more effective at reversing Dex by itself (p < 0.0001). Without reversal, rats took 3584 ± 903.9 (mean ± SD) seconds to emerge from Dex and sevoflurane anesthesia (Fig 6). Atipamezole and caffeine was equally effective at accelerating emergence in male rats receiving Dex supplemented with sevoflurane, taking a mere 69.9 ± 23 sec (S3 Fig) for emergence to take place.

The combination of Dex and low-dose sevoflurane used to generate S3 Table and Fig 6 in female rats was associated with a statistically significant drop in heart rate and respiratory rate, while blood oxygen saturation remained unchanged (Fig 7). Vital signs recorded in Dex by itself were not different from those recorded in Dex with sevoflurane. Male rats were similar (S4 Fig).

## Abdominal surgery

To assess whether Dex in combination with a low dose of another anesthetic could provide robust general anesthesia, we performed abdominal surgery involving incision of the skin, underlying abdominal muscles and the peritoneum on groups of 4 rats (see Fig 8). Immobility

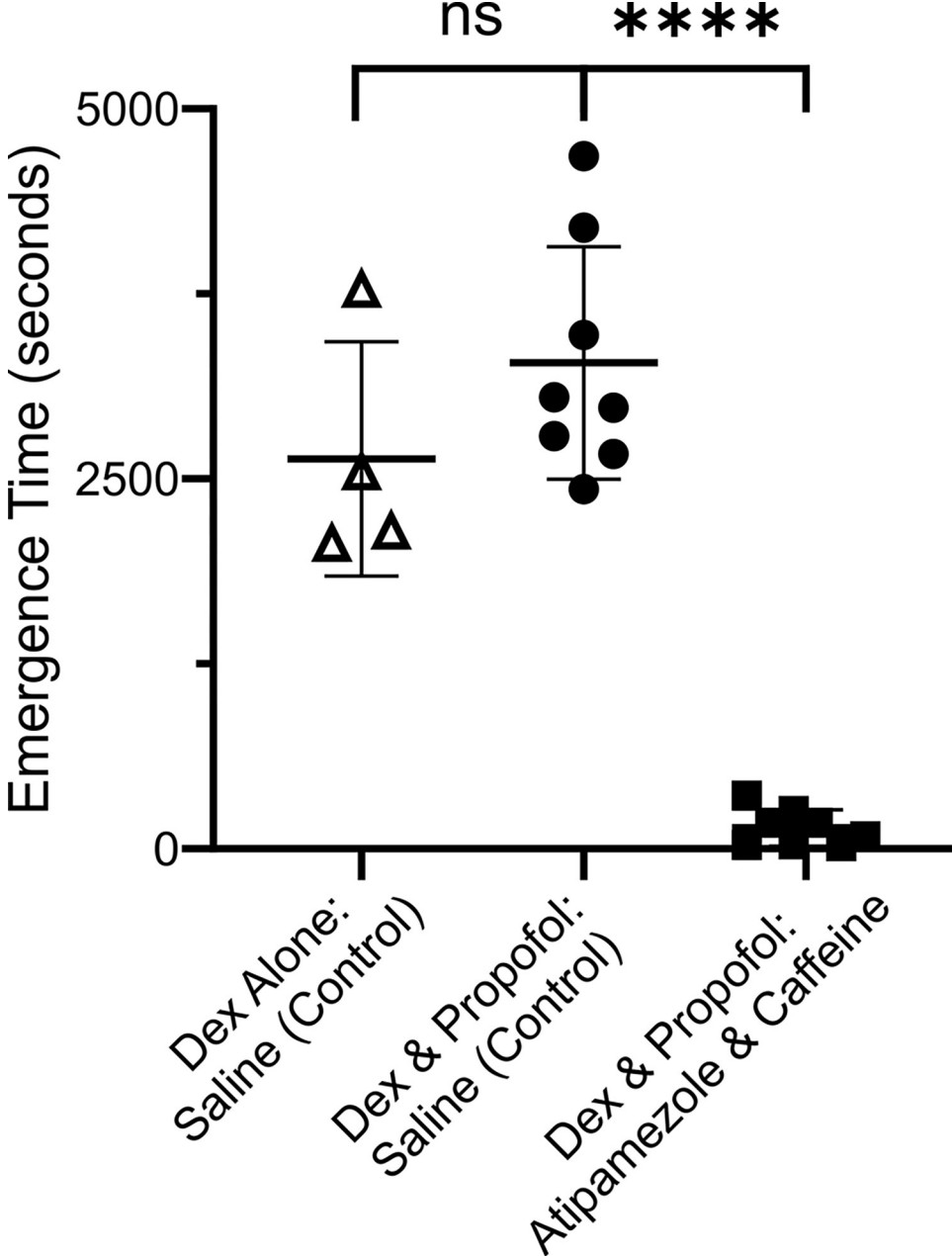

**Fig 4. The combination of atipamezole and caffeine dramatically accelerated emergence from the anesthesia produced by Dex supplemented with a low dose of propofol.** The same group of 8 rats were exposed to two anesthesia sessions, a week apart. At the end of one session the rats received a bolus injection of saline and in the other atipamezole (20 μg/kg) and caffeine (25 mg/kg). The order of the drug injections was randomized. Rats were placed on their backs in a waking box, and the time for the rats to recover their righting reflex was recorded. This time is plotted in the Fig as the Emergence Time. The Fig plots the time to emerge from anesthesia for rats receiving saline (leftmost group) or the same rats receiving atipamezole and caffeine (rightmost group). There was ~95% decrease in Emergence Time. The difference was significant (p < 0.0001, two tailed paired T-test, t = 11.89 and df = 7). Plotted are each data point, the mean value ± standard deviation.

during surgery represents the gold standard for successful anesthesia [26]. These rats were anesthetized with either Dex in combination with 1.4% sevoflurane, or Dex in combination with propofol (4 mg/kg bolus followed by continuous infusion of 300 μg/kg/min). None of the

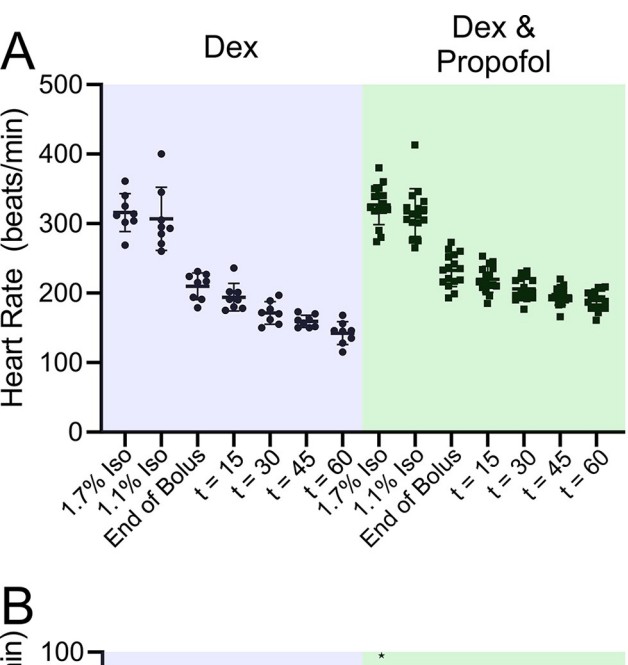

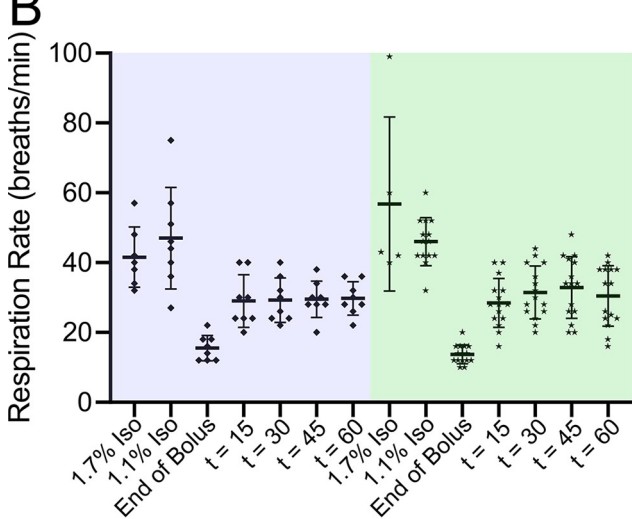

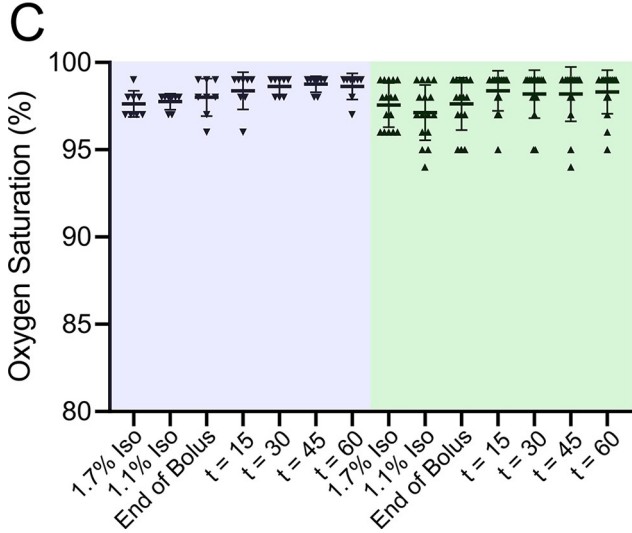

**Fig 5. Female rats exposed to Dex supplemented with a low dose of propofol exhibited slowed heart and respiration rates while blood oxygen saturation levels were unaffected.** The vital signs were measured at various times during the experiment. The first measurement ("1.7% isoflurane") took place while the rats were receiving 1.7% isoflurane, prior to Dex administration. The second measurement ("1.1% isoflurane") was taken while the rats were receiving 1.1% isoflurane, prior to Dex administration. The third measurement ("End of Bolus") was taken immediately after the bolus of Dex was administered. The 1.1% isoflurane was turned off but not yet washed out. Rats were breathing $O_2$/Air. The fourth measurement ("t = 15") was taken fifteen minutes after the end of the bolus. All isoflurane should have been washed out since rats had been breathing $O_2$/Air for 15 minutes. The fifth, sixth and seventh measurements ("t = 30", "t = 45", "t = 60") were taken thirty, forty-five and sixty minutes after the end of the bolus. Statistics: A one-way repeated measures ANOVA was performed to compare the effects of Dex at different time points with that before Dex application. Tukey's HSD Test for Multiple Comparisons found that the mean values for HR were significantly different in the presence and absence of Dex. Comparisons of HR: 1.7% isoflurane vs. 1.1% isoflurane, p = 0.44: We compared the HR at 1.1% isoflurane to the rest of the time points, 1.1% isoflurane vs. End of Bolus, p <0.0001: 1.1% isoflurane vs. t = 15, p <0.0001: 1.1% isoflurane vs. t = 30, p <0.0001: 1.1% isoflurane vs. t = 45, p < 0.0001: 1.1% isoflurane vs. t = 60, p <0.0001. Comparisons of RR: 1.1% isoflurane vs. End of Bolus, p <0.0001: 1.1% isoflurane vs. t = 15, p <0.0001: 1.1% isoflurane vs. t = 30, p <0.0001: 1.1% isoflurane vs. t = 45, p <0.0001: 1.1% isoflurane vs. t = 60, p <0.0001. Comparisons of $SpO_2$: 1.7% isoflurane vs. 1.1% isoflurane, p = 0.95: We compared the HR at 1.1% isoflurane to the rest of the time points, 1.1% isoflurane vs. t = 15, p <0.06: 1.1% isoflurane vs. t = 30, p = 0.41: 1.1% isoflurane vs. t = 45, p = 0.43: 1.1% isoflurane vs. t = 60, p = 0.28.

animals undergoing this surgical procedure exhibited any motor or autonomic response. Fig 8A showed an illustration of the abdominal incision in one rat. Fig 8B plots heart rate, respiratory rate, $SpO_2$ and mean arterial pressure immediately before and after skin incision for Dex and propofol anesthesia. Vital signs were unchanged by incision.

Taken together, the results of these experiments suggest that both combinations tested, Dex with propofol or Dex with sevoflurane, represent potent anesthetics, producing deep levels of unconsciousness, immobility and antinociception.

## EEG analysis: Sevoflurane vs Dex with low sevoflurane, isoflurane vs Dex with low propofol

Dex sdoes not produce a reliable amnestic effect when used by itself at the cllinical doses [47]. Sevoflurane and isoflurane, by contrast, manifest strong amnestic effects with an extremely low incidence of intraoperative awareness or recall in humans [48, 49]. We recorded EEG activities near 1 MAC of sevoflurane or isoflurane, conditions in which awareness and recall are very unlikely, to serve as a baseline for comparison to Dex infusion with low dose sevoflurane (<0.5 MAC) or low dose propofol (300 μg/kg/min). Fig 9A and 9B provide an example of a raw EEG trace and 5-minute spectrogram obtained from a rat anesthetized with 3% sevoflurane. Fig 9C and 9D depicts the raw EEG trace and 5-minute spectrogram obtained in the same rat receiving 1.4% sevoflurane with Dex (12 μg/kg/hr).

Fig 9E compares the power spectra from two 5-minute epochs; one in 3% sevoflurane and the other in 1.4% sevoflurane with Dex (12 mg/kg/hr) averaged from a group of 8 rats. Under both anesthetic conditions, delta bands (0.5–4 Hz) predominated in the power spectrum and were not different between the two periods (p = 0.87, n = 8). The powers in frequency bands of theta (4–8 Hz), alpha (8–12 Hz), spindle (12–15 Hz), and beta (15–25 Hz) were higher under sevoflurane 3% than under sevoflurane 1.4% with Dex (p = 0.003 **, p = 0.002 **, p< 0.000 and p = 0.0008 **, respectively).

Fig 9F plots the burst-suppression ratio over two 5-minute periods, one under sevoflurane 3% (Mean ± SD, 0.79 ± 0.01, n = 8) and the other under sevoflurane 1.4% with Dex infusion (Mean ± SD, 0.18 ± 0.10, n = 8). Sevoflurane 3% produced a higher BSR than did sevoflurane 1.4% with Dex (p <0.0001, n = 8).

Fig 10A and 10B provide an example of a raw EEG trace and 5-minute spectrogram during 1.7% isoflurane anesthesia from one rat. Fig 10C and 10D depicts the raw EEG trace and

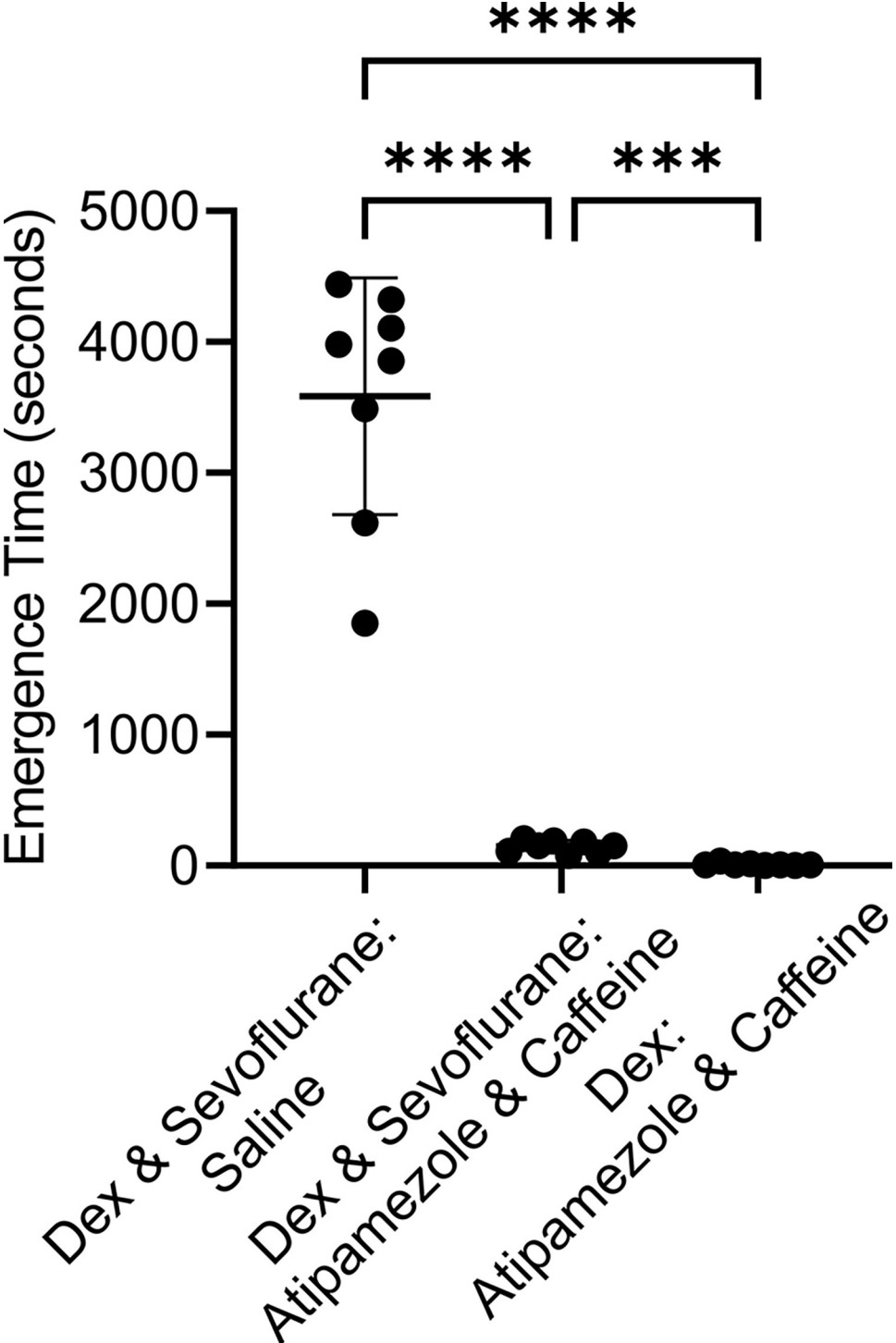

**Fig 6. The combination of atipamezole and caffeine dramatically accelerated emergence from the anesthesia produced by Dex supplemented with a low dose of sevoflurane.** The same group of 8 rats were exposed to two sedation sessions, a week apart. In one anesthesia session the rats received Dex alone while in the second session the rats received Dex and sevoflurane (see Results for details). The order of the sessions was randomized. At the end of both sessions the rats received a bolus injection of atipamezole (10 μg/kg) and caffeine (25 mg/kg). Emergence from anesthesia was fast in both cases, but Dex alone was significantly faster. The difference was significant (p < 0.0001), two tailed paired T-test, t = 7.93 and df = 7). Plotted are each data point, the mean value ± standard deviation.

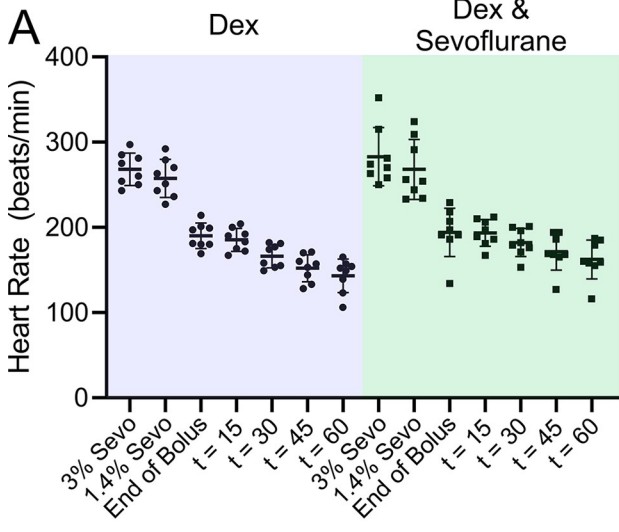

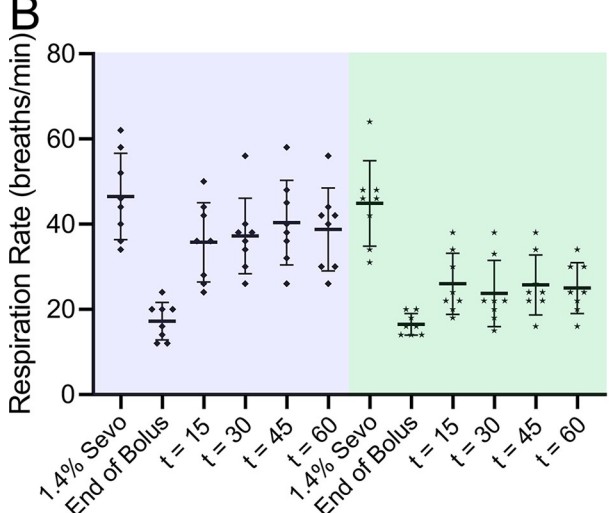

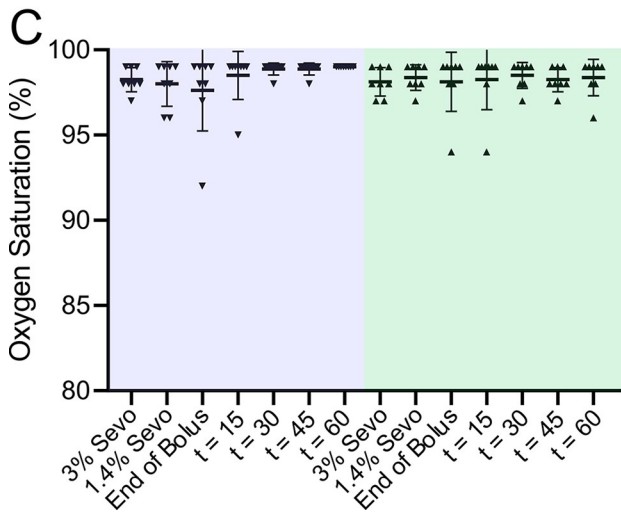

**Fig 7. Rats exposed to Dex supplemented with a low dose of sevoflurane exhibited slowed heart and respiration rates while leaving blood oxygen saturation unaffected.** The vital signs were measured at various times during the experiment. The first measurement took place while the rats were receiving ("3.0% sevoflurane"), prior to Dex administration. The second measurement ("1.4% sevoflurane") was taken while the rats were receiving 1.4% sevoflurane, prior to Dex administration. The third measurement ("End of Bolus") was taken immediately after the bolus of Dex was administered. The 1.4% isoflurane was turned off but not yet washed out. Rats were breathing $O_2$. The fourth measurement ("t = 15") was taken fifteen minutes after the end of the bolus. All sevoflurane should have been washed out since rats had been breathing $O_2$ for 15 minutes. The fifth, sixth and seventh measurements ("t = 30", "t = 45", "t = 60") were taken thirty, forty-five and sixty minutes after the end of the bolus. Statistics: A one-way repeated measures ANOVA was performed to compare the effects of Dex at various times. Tukey's HSD Test for Multiple Comparisons found that the mean values for HR were significantly different in the presence and absence of Dex. Comparisons of HR: 3.0% sevoflurane vs. 1.4% sevoflurane, p = 0.21: We compared the HR at 1.4% sevoflurane to the rest of the time points, 1.4% sevoflurane vs. End of Bolus, p = 0.0012: 1.4% sevoflurane vs. t = 15, p = 0.0003: 1.4% sevoflurane vs. t = 30, p = 0.0001: 1.4% sevoflurane vs. t = 45, p < 0.0001: 1.4% sevoflurane vs. t = 60, p = 0.0001. Comparisons of RR: 1.4% sevoflurane vs. End of Bolus, p = 0.001: 1.4% sevoflurane vs. t = 15, p = 0.053: 1.4% sevoflurane vs. t = 30, p = 0.03: 1.4% sevoflurane vs. t = 45, p = 0.03: 1.4% sevoflurane vs. t = 60, p = 0.019. Comparisons of $SpO_2$: 3.0% sevoflurane vs. 1.4% sevoflurane, p = 0.98: We compared the HR at 1.4% sevoflurane to the rest of the time points, 1.4% sevoflurane vs. End of Bolus, = 0.99: 1.4% sevoflurane vs. t = 15, p >0.99: 1.4% sevoflurane vs. t = 30, p = 0.99: 1.4% sevoflurane vs. t = 45, p = 0.99: 1.4% sevoflurane vs. t = 60, p >0.99.

5-minute spectrogram from the same rat receiving Dex 12 μg/kg/hr with propofol 300 μg/kg/min infusion.

Fig 10E compares power spectra of 5-minute epochs; one taken from 1.7% isoflurane and propofol 300 μg/kg/min with Dex 12 mg/kg/hr infusion from a group of 8 rats. Under both anesthetic conditions, the delta bands (0.5–4 Hz) accounted for the bulk of the power and were not different between the two periods (p = 0.78, n = 8). Alpha bands (8–12 Hz) were the same with both anesthetic conditions (p = 0.32). The powers in theta (4–8 Hz), spindle (12–15), and beta (15–25) frequency bands were higher under isoflurane 1.7% than under propofol with Dex (p = 0.017, p< 0.0001 **** and p = 0.0002 ***, respectively).

Fig 10F plots the burst suppression ratio over two 5-minute periods, one under 1.7% iso-flurane (Mean ± SD, 0.57 ± 0.04, n = 8) and the other under propofol 300 μg/kg/min with Dex 12 mg/kg/hr infusion (Mean ± SD, 0.02 ± 0.01, n = 8). Isoflurane 1.7% caused a higher burst suppression ratio than did low dose propofol with Dex (p<0.0001, n = 8).

No adverse events occurred in the rats throughout this study.

## Discussion

Dex is associated with less postoperative delirium and neurocognitive dysfunction in the elderly [50, 51]. Dex is also associated with less neuroapoptosis and cognitive alterations in developing brains of various animal models, including non–human primates [24, 27, 29, 52–54]. Dex is common in pediatric sedation and as an adjunctive agent in general anesthesia. It is reasonable to posit that using Dex as the primary agent in an anesthetic regimen would maximize its beneficial effects. Unfortunately, Dex is neither a powerful immobilizer nor an amnestic agent. Dex has other drawbacks including prolonged unconsciousness following sedation as well as bradycardia and hypotension. Bradycardia leading to hypotension is usually overcome by fluid bolus in pediatric sedation or fluid and glycopyrrolate in adults. Without a reversal agent available, prolonged recovery times are common.

Atipamezole is a selective $\alpha_2$ receptor antagonist. While it has been shown to reverse the effects of Dex in humans, high doses (Atipamezole: Dex ratio of 40–100:1) were required, which were associated with unwanted effects including emesis, motor restlessness, and increased blood pressure (>20 mm Hg) [55–58]. Due to this unfavorable side effect profile, atipamezole has not been approved for use in humans. For veterinary medicine, the manufacturer recommends an Atipamezole: Dex ratio of 10:1 for rapid reversal [59]. A study in rats

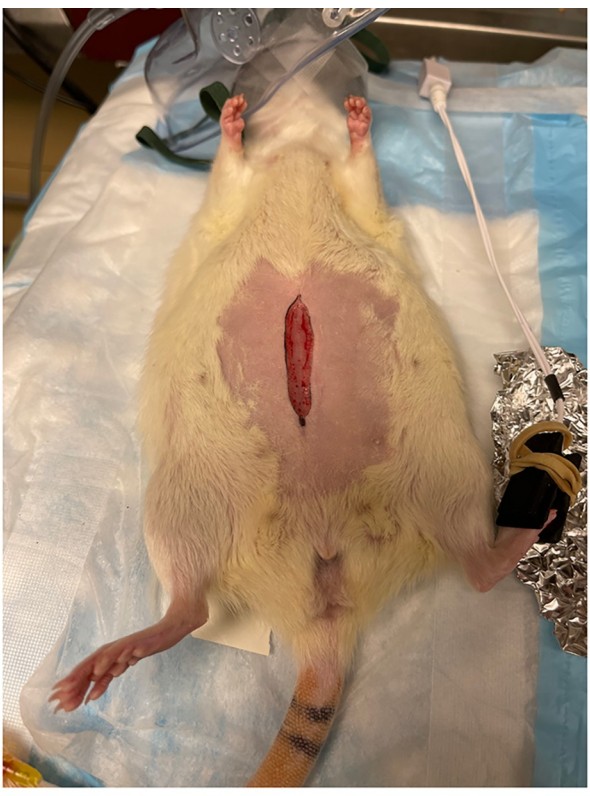

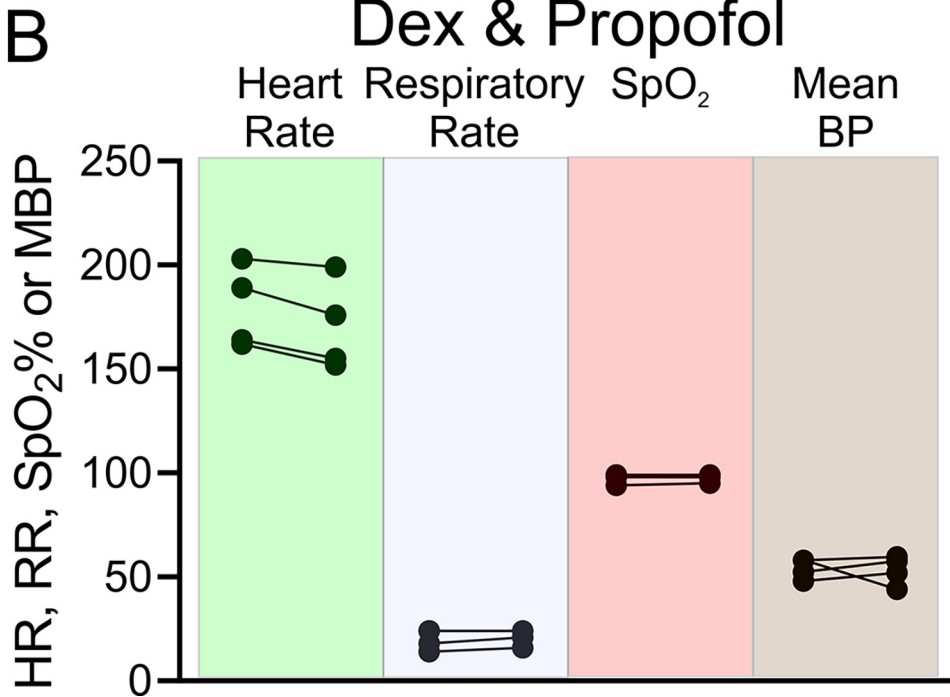

**Fig 8. No change in vital signs observed during surgery for the combination of Dex and low-dose propofol.** A, shows the surgery that took place in one rat. First there was an incision through the skin. Then the abdominal muscles were cut. Finally, the peritoneum was perforated exposing the abdominal cavity. At no stage of the surgery did the rats

respond by moving or by a change in vital signs. B, the combination of Dex and low-dose propofol produced a powerful anesthetic that prevented movement or change in vital signs while surgery was performed. A bolus of Dex (10 μg/kg) was followed by continuous infusion of Dex (12 μg/kg/hr). Propofol was applied as a 4 mg/kg bolus followed by a continuous infusion of 300 μg/kg/min. The skin was cut ~18 minutes after finishing the bolus. Plotted are pairs of measurements taken just before skin incision and just after. A line connects the two points. Heart rate, respiration rate, $SpO_2$ and mean arterial pressure are plotted for each rat, in this group of 4.

from our lab paired low dose atipamezole with caffeine; atipamezole was used at a dose too low to engender adverse effects (atipamezole: Dex ratio 1:1) [35]. Together, the atipamezole with caffeine were remarkably effective and emergence times decreased by ~97% compared to control [35]. Low dose atipamezole and caffeine may represent a clinically useful reversal cocktail for Dex based anesthetics. The absence of such a reversal cocktail has thus far limited the wider use of dexmedetomidine.

In this study, we first evaluated high dose Dex to see if it could produce unconsciousness and immobility. Next, we assessed whether Dex supplemented with a low dose of propofol, or sevoflurane produced unconsciousness and immobility. The secondary agents were used at dosages too low to produce anesthesia or even unconsciousness by themselves. Reversal of anesthesia by atipamezole and caffeine was assessed.

We administered a 10 μg/kg bolus of Dex, followed by either 12 μg/kg/hr or 15 μg/kg/hr infusion for 60 minutes. All rats required ~1 hour to recover with no reversal agent. Dex induced unconsciousness with 100% efficiency but did not reliably produce immobility when tested with a tail clamp, suggesting that these doses of Dex did not produce anesthesia. Increasing the dose of Dex (40 μg/kg bolus, 48 μg/kg/hr infusion for 60 minutes) almost completely suppressed the motor response to the tail clamp (30/32 tests) but also prolonged emergence time by ~3-fold. Interestingly, high dose Dex did not engender further depression of respiratory rate or heart rate beyond that produced by lower doses of Dex, suggesting that the hemodynamic effects saturate. High dose Dex was reversed by atipamezole and caffeine (>99% reduction in emergence time). However, the rats remained sluggish with minimal movement for another 30 to 60 minutes after initial righting, suggesting that high dose Dex is not completely reversible. These results are consistent with reports of surgery carried out with high dose Dex alone in human patients [60].

Previous studies have shown that when Dex was used as an adjunct agent, it reduced the MAC concentration of isoflurane, sevoflurane and other volatile agents in rats and humans [52, 61]. Several clinical trials currently underway are exploring the use of Dex with other anesthetics or sedatives to maximize clinical benefit and minimize side effects [61–64]. Dex was used as an adjunct at doses ranging from 0.3–0.7 mcg/kg/hr in these studies. At these doses, Dex alone is not able to cause unconsciousness in humans subjects, but can reduce the MAC concentration of sevoflurane by 20–30%. When Dex was used at a higher dose alone, it prolonged recovery significantly [60]. Without a clinically effective reversal for Dex, the use of higher doses of Dex with or without a second agent is not practical. With the success of using low dose atipamezole and caffeine combination to reverse Dex effectively,we are able to employ anesthetic combinations in which Dex serves as the primary agent, with only small subanesthetic doses of another agent as an adjunct to enable Dex to function as a robust general anesthetic. We characterize Dex as the "primary" agent in our anesthetic combinations because it is employed at a dose that produces greater "anesthetizing power" than the dose of the other agent with which we combine it. This dex dose alone is able to maintain unconsciousness in rats throughout the duration of its infusion, while both the low dose sevoflurane and propofol doses we used in our combination regimens are unable to do so when used alone. Our anesthetic combinations represent an alternative and corollary approach that

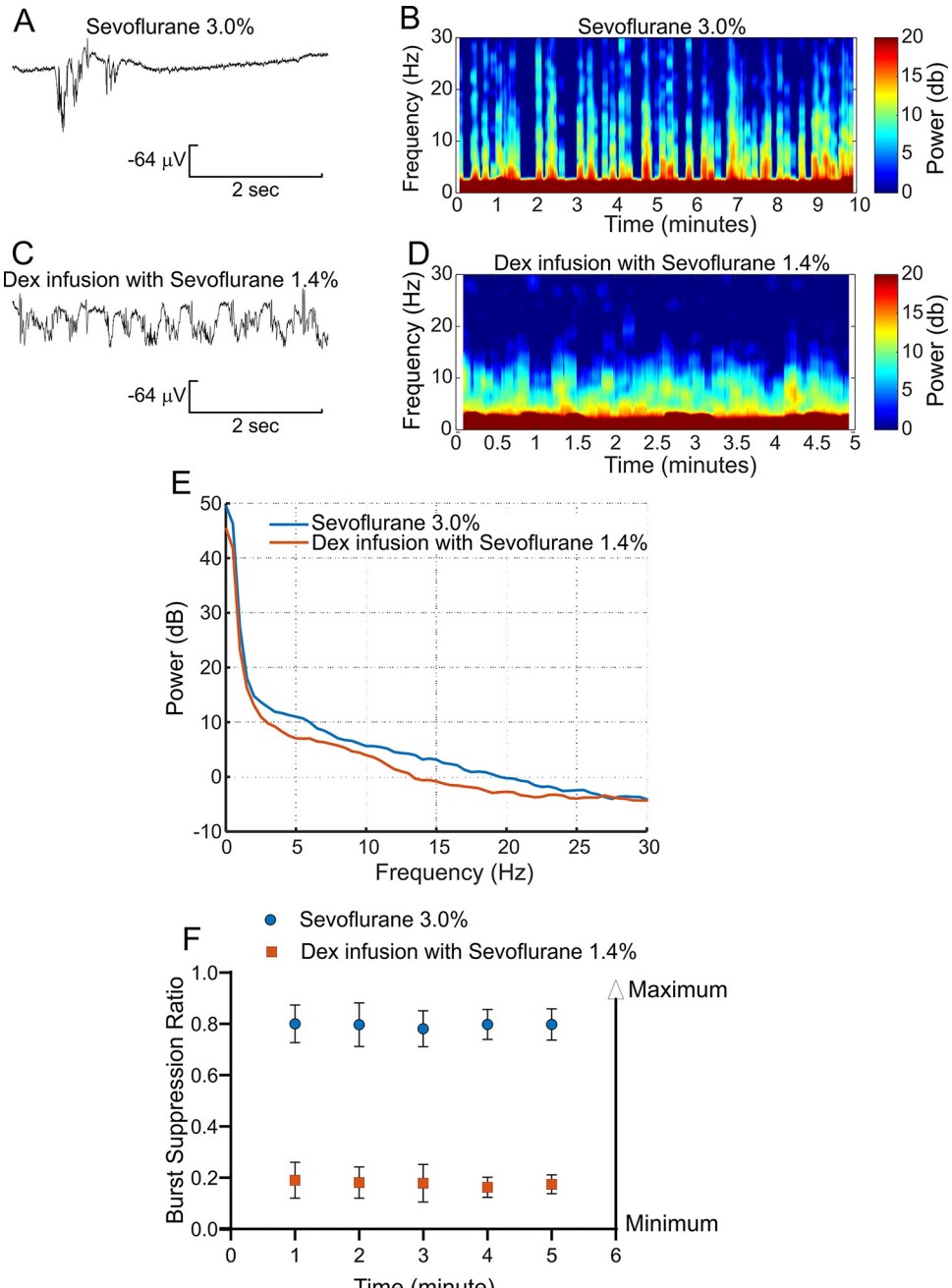

**Fig 9. Comparing EEGs obtained under 3% sevoflurane and under Dex with low dose sevoflurane.** EEG recordings are obtained from the anterior lead. A and B are representatives of a raw EEG trace and a 5-minute spectrogram recorded from a period under 3% sevoflurane anesthesia from one rat. C and D are representatives of a raw EEG trace and 5-minute spectrograms recorded from a period under sevoflurane 1.4% with Dex (12 μg/kg/hr) from the same rat in A and B. E, Comparison of two 5-minute epoch power spectra, one under sevoflurane 3% and another under sevoflurane 1.4% with Dex (12 μg/kg/hr) obtained from 8 rats. Powers (dB) frequency bands between sevoflurane 3.0% and sevoflurane 1.4% with Dex are shown as delta (0.5–4 Hz) p = 0.87, 19.60 ± 10.71 vs 18.60 ± 13.32; theta (4–8 Hz) p = 0.003 **, 10.31 ± 1.79 vs 7.72 ± 0.95; alpha (8–12 Hz) p = 0.002 **, 6.63 ± 0.76 vs 4.46 ± 1.41; spindle (12–15 Hz) p< 0.0001 **** 4.53 ± 0.66 vs 1.10 ± 0.92; and beta (15–25 Hz) p = 0.0008 ***, 0.91 ± 1.55 vs -1.86 ± 0.99, n = 8). Under both anesthetic conditions, delta bands were the dominant power and not different between the two periods. Frequency bands of theta, alpha, spindle, and beta were higher under sevoflurane 3%than under sevoflurane 1.4% with Dex. The unpaired-T test was used to compare the frequency bands between two anesthesia conditions. F, The burst-suppression ratio (BSR) over two 5-minute periods, one under sevoflurane 3% (Mean ± SD, 0.79 ± 0.01, n = 8) and the other under sevoflurane 1.4% with Dex infusion (Mean ± SD, 0.18 ± 0.10, n = 8). Sevoflurane 3% produced higher BSR than sevoflurane 1.4% with Dex. Burst suppression ratio (BSR) in the EEG was calculated by a formula, BSR = (total

time of suppression/epoch length) x 100% and analyzed independently by two different members of this study. Each independent analysis produced consistent results. Suppression time was defined from 0.5 to 5 seconds. EEG suppression was defined as an amplitude $< 5 \, \mu V$ which lasted for $\geq 30\%$ of each minute.

emphasizes Dex as the central component and employs a minimal dose of a second agent in an adjunctive capacity. With this strategy, the Dex doses evaluated in our study are nevertheless closer to the doses used in humans than the ones used in Veterinary Medicine. Dex doses up to 3 $\mu$g/kg/hr are commonly used for pediatric procedural sedation, such as that used for MRI scans [65]. In contrast, Dex 0.5–0.75 mg/kg IP is used in combination with ketamine at 75–150 mg/kg IP to anesthetize small animals, like rats, for surgery (see for example - https://animalcare.ubc.ca/sites/default/files/documents/Guideline%20-%20Rodent%20Anesthesia%20Analgesia%20Formulary%20%282016%29.pdf).

Infusing propofol (300 $\mu$g/kg/min) in combination with Dex (12 $\mu$g/kg/hr) completely suppressed any responses to the tail clamp and surgery in all animals tested. Dex and propofol together appear to recapitulate all aspects of an effective general anesthetic. Moreover, the unconsciousness produced by Dex and propofol was rapidly reversed by low dose atipamezole with caffeine (Fig 4).

Combining Dex with 1.4% sevoflurane (<0.5 MAC) produced unconsciousness and complete immobility in response to tail clamp and surgery. This drug combination was also rapidly reversed by a low dose of atipamezole with caffeine (Fig 6).

The combination of atipamezole and caffeine is as effective for reversing Dex with propofol or Dex with sevoflurane as it is in reversing Dex by itself [35]. This effective reversal provides a solution for the slow emergence of Dex based anesthetics. This may increase the potential for clinical use of Dex based anesthetics.

While it is difficult to directly assess awareness or amnesia in rats, the question can be addressed indirectly. Firstly, while Dex is not an effective amnestic when used alone, the agents we paired them with have been shown to be highly effective amnestic agents on their own at the dosages we employed [33, 66, 67]. Secondly, the lack of motor response to the tail clamp as well as trans-peritoneal abdominal incision suggests that an adequate depth of anesthesia was achieved. Finally, data obtained from EEG recordings of our anesthetized animals suggested that memory was impaired based on the similarities of EEG patterns produced by ~ 1 MAC sevoflurane or isoflurane and our Dex combination regimens. We compared the EEG activity recorded under low dose sevoflurane with Dex or low dose propofol with Dex to 1 MAC of sevoflurane or isoflurane alone. With all these anesthetic regimens, the power spectra were similar, characterized by dominance of delta frequency bands. Slow waves (delta band) are consistently present in the surgical phase of general anesthesia [68]. Notably, low dose sevoflurane with Dex and low dose propofol with Dex produced very low burst suppression ratios while sevoflurane or isoflurane near MAC levels produced much higher burst suppression ratios. Studies suggest that prolonged burst suppression under general anesthesia may be associated with higher incidences of post-operative delirium and neurocognitive dysfunction in elderly patients [69]. Therefore, avoiding lengthy periods of burst suppression may benefit vulnerable populations.

Using Dex as a primary anesthetic supplemented with a subanesthetic dose of second agent may represent a more receptor specific anesthetic strategy which limits unwanted pleiotropic effects. Inhalational agents produce their anesthetic effects, unconsciousness and immobility, by interacting with a variety of receptors, ion channels and second messenger systems within neural circuits [26, 70]. The sites of anesthetic actions of unconsciousness and immobility are in the brain and spinal cord, respectively. Despite years of research, a comprehensive mechanistic understanding of the anesthetic state remains elusive [71]. In vitro, inhalational or

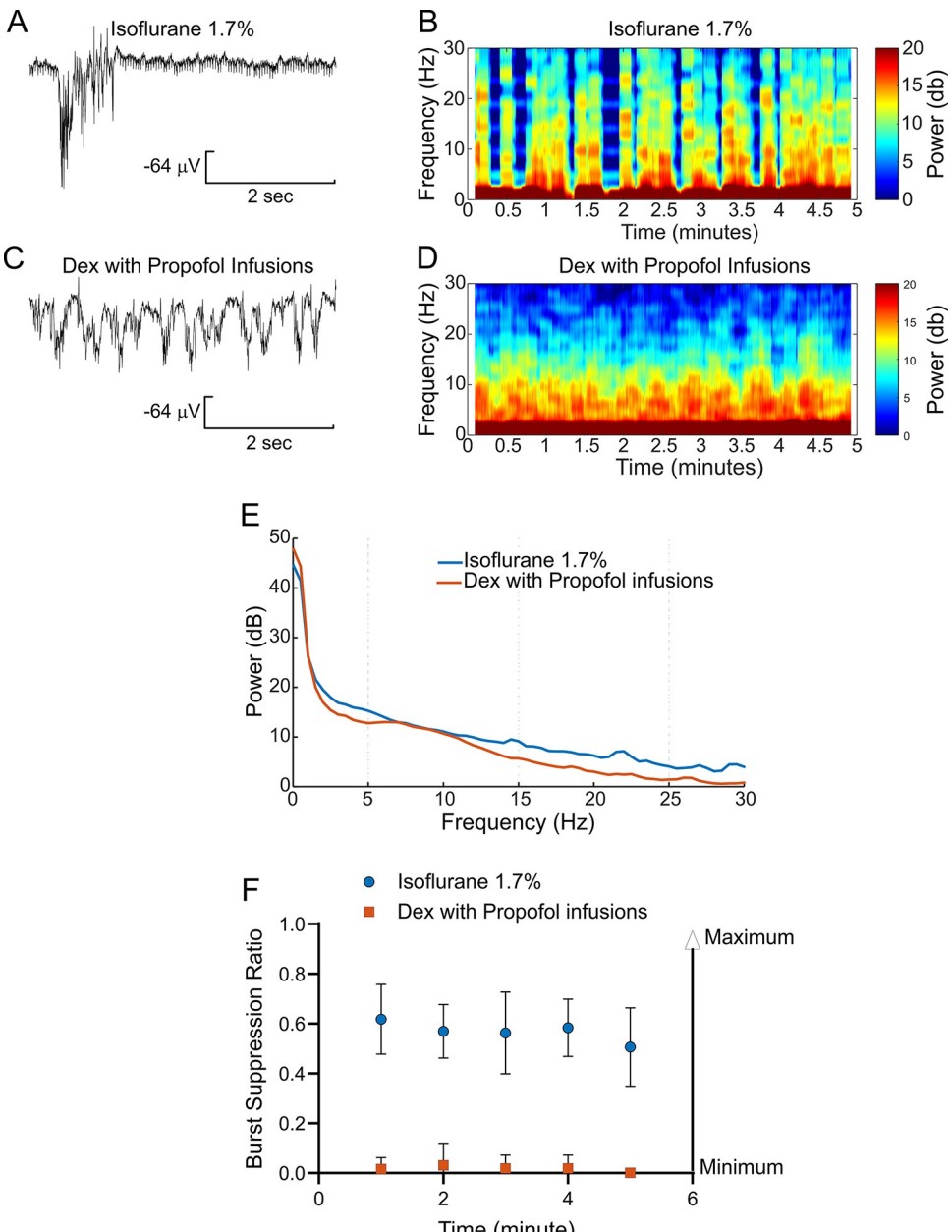

**Fig 10. Comparisons EEGs obtained under isoflurane 1.7% and Dex with low dose propofol (300 µg/kg/min).** A and B are representatives of a raw EEG trace and a 5-minute spectrogram recorded from a period of 1.7% isoflurane anesthesia from one rat. C and D are representatives of a raw EEG trace and a 5-minute spectrograms recorded from a period of anesthesia under low dose propofol (300 µg/kg/min) with Dex (12 µg/kg/hr) infusion from the same rat in A and B. E, Comparison of two 5-minute epoch power spectra, one under isoflurane 1.7% and another under low dose propofol (300 µg/kg/min) with Dex (12 µg/kg/hr) from 8 rats. Powers of frequency bands between isoflurane 1.7% and low dose propofol with Dex are shown as delta (0.5–4 Hz) p = 0.78 21.99 ± 8.54 vs 20.66 ± 10.48, theta (4-8Hz) p = 0.017 * 14.12 ± 1.34 vs 12.86 ± 0.38, alpha (8–12 Hz) p = 0.32 11.06 ± 0.82 vs 10.51 ± 1.29, spindle (12–15 Hz) p< 0.0001 **** 9.31 ± 0.36 vs 6.81 ± 1.03 and beta (15–25 Hz) p = 0.0002 *** 6.50 ± 1.32 vs 3.20 ± 1.32. Under both anesthetic conditions, delta bands were the dominant power and not different between the two periods. Alpha bands were similar under both anesthetic conditions. Frequency bands of theta, spindle and beta were higher under isoflurane 1.7% than under propofol with Dex. The unpaired-T test was used to compare the frequency bands between anesthesia conditions. F, The burst-suppression ratio (BSR) over two 5-minute periods, one under 1.7% isoflurane (Mean ± SD, 0.57 ± 0.04, n = 8) and the other under propofol 300 µg/kg/min with Dex 12 mg/kg/hr infusion (Mean ± SD, 0.02 ± 0.01, n = 8). Isoflurane 1.7% caused higher BSR than low dose propofol with Dex. Burst suppression ratio (BSR) in the EEG was calculated by a formula, BSR = (total time of suppression/epoch length) x

100% and analyzed independently by two different members of this study. Each independent analysis produced consistent results. Suppression time was defined from 0.5 to 5 seconds. EEG suppression was defined as an amplitude $< 5\ \mu V$ which lasted for $\geq 30\%$ of each minute.

intravenous anesthetic agents produce their effects at high concentrations [72]. At these high concentrations, anesthetics interact promiscuously with multiple targets and may result in untoward effects including nausea, respiratory depression, myocardial suppression, and immunomodulation.

By contrast, Dex is the most selective agent currently used for either sedation or anesthesia, exerting its effects at nanomolar concentrations in vitro [73, 74]. Dex selectively activates $\alpha_2$ receptors, which activates the $G_{i/o}$ signaling pathway resulting in the inhibition of adenylate cyclase, thereby lowering intracellular cAMP levels [75, 76]. Activation of this pathway also activates GIRK-$K^+$ channels and inhibits voltage gated $Ca^{2+}$ channels. Activation of the $G_{i/o}$ pathway inhibits neuronal activity and decreases neurotransmitter release.

Caffeine increases $[cAMP]_i$ levels by inhibiting phosphodiesterase, which breaks down cAMP, thereby directly countering an important cellular effect of Dex. We predict that other stimulants which elevate $[cAMP]_i$ should be equally effective. Low dose atipamezole and caffeine seem to work synergistically to reverse Dex mediated sedation.

A recent study showed that a large dose of amphetamine, a potent stimulant with a half-life of ~10 hours, could rapidly reverse Dex by itself [77]. A combination of low dose atipamezole with a lower dose of amphetamine may be able to reverse dexmedetomidine rapidly while minimizing the prolonged effects of the stimulant.

The ability to rapidly reverse sedation engendered by even high doses of dexmedetomidine overcomes one of the major barriers to its routine application in general anesthetic regimens. This study represents an incremental advance toward realizing the goal of receptor targeted anesthesia articulated by Talke in his 1998 editorial [78]. Our goal is to create Dex based anesthetic combinations that are even more specifically receptor targeted, and to extend these strategies into clinical studies.

## Limitations

Plasma concentrations of Dex were not measured. Human studies are needed to correlate the Dex concentrations in plasma to the anesthetic effects. In these studies, EEG leads were placed after rats were already anesthetized. Therefore, there was no baseline EEG before anesthesia. The drug combinations demonstrated strong antinociceptive effects during surgeries. Whether Dex supplemented with a second agent will reduce the use of opioids intraoperatively and minimize side effects postoperatively remains to be determined. Which of the anesthetic combinations proves to be "safest" in terms of neuroapoptosis in neonatal animal studies or in ameliorating cognitive decline or emergence delirium in the elderly remails to be determined. Studies in "aged" rats will provide useful insights into the safety of the Dex based drug combinations.

In summary, our study demonstrates that Dex supplemented with a low dose of either propofol or sevoflurane creates a potent anesthetic with a favorable safety profile that can be rapidly reversed by low dose atipamezole with caffeine. Translating these observations to human populations represents a high priority.

## Supporting information

**S1 Checklist. The ARRIVE guidelines 2.0: Author checklist.**
(PDF)

**S1 Fig. Scalp EEG lead placement.** Two scalp electrodes were placed, as shown. We drew a line between the anterior edge of bilateral ears, between Bregma and Lambda. From the mid-point, one electrode was placed anteriorly perpendicular to the line and the other posteriorly perpendicular to it. Two EEG channels were recorded, the first one (red) from an electrode placed over the anterior portion of the brain, and a second electrode (green) placed over the posterior portion of the brain. The EMG lead (yellow) was obtained from an electrode placed over the left shoulder, all referenced to an electrode (white) placed near medial to the ears. A ground electrode (black) was placed on the opposite side of the reference lead. Signals recorded from the anterior lead were analyzed for global changes during anesthesia.
(TIF)

**S2 Fig. Determining the dose of propofol required to maintain unconsciousness to define a low dose of the drug.** For this experiment a group of rats received a bolus of 5 mg/kg of propofol, applied in 5 minutes via a pump, after which they received a continuous infusion of propofol, at different concentrations for an additional 60 minutes. All rats remained unconscious for the 60-minute infusion if the infusion rates of propofol were kept at or above 400 μg/kg/min. In contrast, 300 μg/kg/min was not sufficient to keep the rats unconscious during the infusion.
(TIF)

**S3 Fig. The combination of atipamezole and caffeine dramatically accelerated emergence from anesthesia produced by Dex alone or from the combinations of Dex with Sevoflurane or Dex with Propofol, in male rats.** The same group of 8 rats were exposed to three sedation sessions, a week apart. At the end of each session the rats received a bolus injection of atipamezole (10 μg/kg) and caffeine (25 mg/kg). Rats were placed on their backs in a waking box, and the time for the rats to recover their righting reflex was recorded. Data (RORR Times in seconds): Dex alone—5, 9, 5, 1, 29, 1, 15, 2, Dex with Sevoflurane—110, 72, 81, 59, 62, 32, 85, 59, Dex with Propofol—222, 182, 150, 44, 32, 59, 129, 25.
(TIF)

**S4 Fig. Male rats exposed to Dex alone or supplemented with a low dose of propofol or Dex supplemented with a low dose of sevoflurane exhibited slowed heart and respiration rates while leaving blood oxygen saturation unaffected.** Dex or Dex supplemented with either propofol or sevoflurane was applied at time = 0. Vital signs were then measured every 15 minutes. Comparisons of HR at different times for Dex 10/12. Dex/sevoflurane and Dex/propofol were similar and are not presented. For this analysis a repeated measures two-way ANOVA was employed: 1.7% isoflurane vs. 1.1% isoflurane, p = ns: We compared the HR at 1.1% isoflurane to the rest of the time points. 1.1% isoflurane vs. End of Bolus, $p < 0.0001$: 1.1% isoflurane vs. t = 15, p = 0.0001: 1.1% isoflurane vs. t = 30, $p < 0.0001$: 1.1% isoflurane vs. t = 45, $p < 0.0001$: 1.1% isoflurane vs. t = 60, $p < 0.0001$, Comparisons of RR: Only the following times were different. 1.1% isoflurane vs. End of Bolus, $p < 0.0001$: End of bolus vs t = 30, p = 0.0006: End of bolus vs t = 45, p = 0.0001: End of bolus vs t = 60, $p < 0.0001$: Comparisons of $SpO_2$: No significant changes were observed.
(TIF)

**S1 Table. Comparing responses to a noxious stimulus in female rats exposed to a lower dose of Dex alone with rats administered a higher dose of Dex alone.** A noxious stimulus was applied at different time points in an experiment. Responses were tabulated and are presented numerically in the table. Statistical difference is calculated with Fisher's exact test.
(TIF)

**S2 Table. Comparing responses to a noxious stimulus in rats exposed to Dex alone with rats administered Dex and a low dose of propofol.** A noxious stimulus was applied at different time points in an experiment. Responses were tabulated and are presented numerically in the table. Statistical difference is calculated with Fisher's exact test.
(TIF)

**S3 Table. Comparing responses to a noxious stimulus in rats exposed to Dex alone with rats administered Dex and a low dose of either sevoflurane or propofol.** A noxious stimulus was applied at different time points in an experiment. Responses were tabulated and are presented numerically in the table. Statistical difference is calculated with Fisher's exact test. A, data from female rats. B, data from male rats.
(TIF)

**S4 Table. A—determining the concentration of sevoflurane required to prevent half of the responses to a noxious mechanical stimulus.** Details about the stimulus are in the Methods. The green column represents ~1 MAC equivalence concentration or $EC_{50}$.
(TIF)

# Acknowledgments

We would like to thank the support from the Department and Anesthesia and Critical Care at the University of Chicago. We thank Chuanhong Liao, MS, the Biostatistics lab, Department of Public Health Science, the University of Chicago, for her help with the statistics. We also thank Dr, Vernon L. Towle, Professor, Department of Neurology, University of Chicago, for his support and advice on EEG recording and analysis.

# Author Contributions

**Conceptualization:** Zheng Xie, Robert Fong, Aaron P. Fox.

**Data curation:** Zheng Xie, Robert Fong, Aaron P. Fox.

**Formal analysis:** Zheng Xie, Robert Fong, Aaron P. Fox.

**Funding acquisition:** Zheng Xie, Aaron P. Fox.

**Investigation:** Zheng Xie, Robert Fong, Aaron P. Fox.

**Methodology:** Zheng Xie, Aaron P. Fox.

**Project administration:** Zheng Xie, Aaron P. Fox.

**Resources:** Zheng Xie, Aaron P. Fox.

**Software:** Zheng Xie, Aaron P. Fox.

**Supervision:** Zheng Xie, Aaron P. Fox.

**Validation:** Zheng Xie, Robert Fong, Aaron P. Fox.

**Visualization:** Zheng Xie, Aaron P. Fox.

**Writing – original draft:** Zheng Xie, Aaron P. Fox.

**Writing – review & editing:** Zheng Xie, Robert Fong, Aaron P. Fox.

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
