## [Decision Letter · Decision Letter 0]

23 May 2023

PONE-D-23-07486Towards A Potent and Rapidly Reversible Dexmedetomidine-Based General AnestheticPLOS ONE

Dear Dr.  Xie,

Thank you for submitting your manuscript to PLOS ONE. After careful consideration, we feel that it has merit but does not fully meet PLOS ONE’s publication criteria as it currently stands. Therefore, we invite you to submit a revised version of the manuscript that addresses the points raised during the review process.

ACADEMIC EDITOR: Please assess carefully all reviewers comments.

We look forward to receiving your revised manuscript.

Kind regards,

Silvia Fiorelli

Academic Editor

PLOS ONE

Journal Requirements:

2. As part of your revision, please complete and submit a copy of the Full ARRIVE 2.0 Guidelines checklist, a document that aims to improve experimental reporting and reproducibility of animal studies for purposes of post-publication data analysis and reproducibility: https://arriveguidelines.org/sites/arrive/files/documents/Author%20Checklist%20-%20Full.pdf. Please include your completed checklist as a Supporting Information file. Note that if your paper is accepted for publication, this checklist will be published as part of your article.

"This study is supported by a NIH grant (GM-116119) To ZX and APF"       

Reviewers' comments:

Reviewer's Responses to Questions

**Comments to the Author**

1. Is the manuscript technically sound, and do the data support the conclusions?

Reviewer #1: Partly

Reviewer #2: Yes

2. Has the statistical analysis been performed appropriately and rigorously? 

Reviewer #1: I Don't Know

Reviewer #2: Yes

3. Have the authors made all data underlying the findings in their manuscript fully available?

Reviewer #1: Yes

Reviewer #2: Yes

4. Is the manuscript presented in an intelligible fashion and written in standard English?

Reviewer #1: No

Reviewer #2: Yes

5. Review Comments to the Author

Reviewer #1: In Manuscript PONE-D-23-07486, Xie and colleagues present their research examining the effect of adding dexmedetomidine (DEX) with other general anesthetics, in order to create a more readily reversible anesthetic. Overall, the authors are to be commended on the amount of data presented in this manuscript, however, this large volume of data obscures issues with the design of their experiments. There are almost three separate papers and hypotheses tested. The experimental design would have been better served if the author concentrated on one general anesthetic (e.g., sevoflurane) and performed more complete experiments. Overall, the results are confusing as presented. It appears that the authors wanted to test whether the addition of DEX to a standard general anesthetic can reduce the dose of general anesthetic required. However, the authors tested four different general anesthetics using a model for anesthetic effects (loss of tail clamp response) and recovery (recovery of righting reflex – RORR). This model was then further examined using a surgical model (abdominal incision).

Due to the large amount of data, their message is confusing to discern. The paper would be better suited for publication if the paper was presented as a single general anesthetic study that was more complete. A subsequent paper could build from these findings, and further examine other general anesthetics. Specific comments are below.

1. The authors wished to examine if a low or high dose of DEX could enhance a general anesthetic’s effects, thus decreasing the dose (MAC) of anesthetic used. The authors first determined the MAC of isoflurane and sevoflurane. Interestingly, the authors found that their MAC was higher than those published previously by others using a rat model and explain the discrepancy as being due to the tail clamp stimulus.

Tables 1A/1B should be moved to Supplemental Data. It is background information that could be moved to decrease the size of the paper.

2. In a similar manner, the authors determined a “low dose” and “high dose” of propofol, but they did not use the tail clamp test, instead they used “consciousness.” Why did they not use the tail clamp test to be more consistent?

3. The data is hard to read as written. For example, page 14 (Paragraph starting with “Table 2”) presents a lot of data with associated p values, and it just doesn’t read well. This issue can be found throughout the results section, and I would suggest that these sections be rewritten without so much data. Also, why did the authors study 200 and also 300 mcg/kg/hr of propofol if 300 mcg/kg/hr was considered to be low dose. In essence, two “low doses” of propofol were studied.

4. The authors also wanted to see if DEX could be reversed using a commonly known and utilized antagonist, atipamezole. This drug has been used extensively in the veterinary community for reversal of DEX or its non-racemic mixture medetomidine. Thus I am not sure why the authors present atipamezole as a new agent. It may not work as well in humans (as they mention in their discussion), but it is not novel in my opinion.

If the authors wish to study reversal of DEX effects, it should not have been included in the same study as examining “low dose” propofol. This idea should have been tested separately.

5. The paper would have been better organized by studying one general anesthetic completely, without reversal. Then, they could have added a section examining DEX reversal from a general anesthetic. Similarly, the abdominal surgery portion of the study, could have been done with a single general anesthetic agent, not four general anesthetics.

6. Low and high dose DEX data (Figure 10 and Table 5) should be presented first, as the showing this effect should have been the initial study done before examining combinations of DEX with general anesthetics.

7. The data on using DEX combined with propofol, isoflurane, and sevoflurane for abdominal surgery does follow their prior data using a tail clamp, both are painful stimuli, but is the lack of change in vital signs proof of a complete anesthetic? This data could be removed without affecting their interpretation.

8. The EEG analysis is quite complex, and it appears that it is included to demonstrate that the DEX/general anesthetic combinations effect memory and awareness. Again, this data could be a separate paper and by combining it with their other data, the resulting message/interpretation is confused. Why did they feel the need to demonstrate these effects?

Again, the combination of Tail clamp experiments with four different general anesthetics, abdominal surgery, and EEG analyses seems unnecessary to demonstrate that DEX can enhance lower dose general anesthetics to produce an “anesthetized” state.

9. The protocols used seem a bit random. For example, rats were anesthetized with isoflurane prior to surgery required for measuring EEG, then DEX was given, and the animals given propofol boluses and then an infusion. Various drugs were given after 30 or 60 minutes (why these timepoints?), infusion doses were changed (e.g., DEX was dropped from 15 mcg/kg/hr to 12 mcg/kg/hr) but why?

10. The Discussion describes a lot of prior work on DEX reversal agents, which doesn’t appear to be the major point of the manuscript and has been previously published by this laboratory (Ref. 42). Again, this focus on DEX reversal takes away from their findings.

11. The authors state that they wish to examine if combining DEX with general anesthetics can allow for “sub-therapeutic” doses of the general anesthetic. In fact, other published work has already demonstrated that DEX can reduce the MAC of volatile agents in animals and humans. If this fact is known, then what does this study add to the literature? They state they their desire is to determine if DEX can be used as the primary anesthetic, but when combining it with other agents, which one is primary and which one is secondary? Or does it matter? I feel that the authors are arguing a fine point that is not as important.

The question is can one reduce the amount of general anesthetic needed (whether it is isoflurane, sevoflurane, propofol, or nitrous oxide) when combined with DEX. Again, this fact has been demonstrated so what does this study add besides a more thorough examination of multiple drug combinations?

12. The authors discuss a major limitation of their study, simply put, they only studied female rats. It has been shown that female rats are more sensitive to DEX effects than male rats. This issue seriously limits the interpretation of their data.

What the authors do not discuss completely in my opinion is why their data demonstrates such a major effect of DEX compared to prior work.

Reviewer #2: This is a feasibility study in rodents to test whether combining dexmedetomidine with low doses of conventional anesthetics is sufficient to provide surgical anesthesia. The rationale for the study is that conventional anesthetics are known to cause delirium and cognitive dysfunction in elderly patients, and dexmedetomidine is known to be less deleterious. The manuscript is well written, and the results are described clearly. However, I have several concerns that need to be addressed.

• The short title, “A strategy for creating a new anesthetic,” is misleading. “New anesthetic” implies a novel drug, but the authors describe a novel dosing regimen using existing anesthetics. This should be revised.

• As stated in the Abstract and Introduction, the premise of the study is that dex administration will allow for lower doses of conventional anesthetics that are associated with delirium and cognitive dysfunction in the elderly. However, the authors did not use aged animals and did not test for delirium or cognitive dysfunction in their study.

• A significant portion of the Introduction discusses the methods, results, and conclusions of the study. This content belongs in the Methods, Results, and Discussion sections, respectively. The Introduction should focus on the background and rationale.

• Why were only female rats used for the study? The NIH and most journals now require the use of both sexes to account for sex as a biological variable.

• Were the anesthetic exposures conducted in random order?

• It should be clearly stated in the manuscript that atipamezole is not approved for human use. This greatly limits the translational potential of these results to the clinical setting.

• There are far too many figures. Many of them should be combined.

• I find it curious that MAC values for sevoflurane and isoflurane were much higher than reported values in the literature. The authors attribute this to their tail clamp being a more potent noxious stimulus, but are other explanations possible? Were the vaporizers and agent analyzers properly calibrated? Maybe the equilibration times were too short?

• In the Discussion, it seems arbitrary to call dex the “primary” anesthetic agent in these studies. The study showed that combining sub-anesthetic doses of dex and conventional anesthetics is sufficient to produce surgical anesthesia, so in my view, neither is the “primary” anesthetic.

6. PLOS authors have the option to publish the peer review history of their article (what does this mean?). If published, this will include your full peer review and any attached files.

Reviewer #1: **Yes: **Timothy Angelotti MD PhD

Reviewer #2: No

---

## [Author Response · Author response to Decision Letter 0]

18 Jul 2023

PONE-D-23-07486

Towards A Potent and Rapidly Reversible Dexmedetomidine-Based General Anesthetic

July 18, 2023

Dear Dr Fiorelli,

We would like to thank the reviewers and the editor for their excellent comments. The changes made in response to the reviews have undoubtedly improved the manuscript. 

This letter includes a point-by-point response to each of the comments made by the reviewers and the editor. The reviewer’s abbreviated comments are in bold italics followed immediately by our reply in regular font. The line numbers in the reply to the reviewers correspond to the line numbers in the version with “Tracked Changes – Simple Markup.”

We have followed the PLOS One style requirements including file naming conventions. 

2. As part of your revision, please complete and submit a copy of the Full ARRIVE 2.0 Guidelines checklist. 

The Full Arrive 2.0 Guidelines Checklist has been included with the manuscript resubmission.

"This study is supported by a NIH grant (GM-116119) To ZX and APF." 

This amended role for the funders is accurate and it has been included in the Cover Letter. 

Reviewers' comments:

Reviewer's Responses to Questions

Comments to the Author

1. Is the manuscript technically sound, and do the data support the conclusions?

Reviewer #1: Partly

Reviewer #2: Yes

The manuscript has been edited to make it technically sound and to have conclusions supported by the data.

2. Has the statistical analysis been performed appropriately and rigorously? 

Reviewer #1: I Don't Know

Reviewer #2: Yes

The statistics have been redone in a manner that will make it easier to follow.

3. Have the authors made all data underlying the findings in their manuscript fully available?

Reviewer #1: Yes

Reviewer #2: Yes

No changes were made to the data underlying the findings with the exception that the data from new figures is now included.

4. Is the manuscript presented in an intelligible fashion and written in standard English?

Reviewer #1: No

Reviewer #2: Yes

The revised manuscript has been comprehensively edited as can be seen by the “Tracked Changes” version. Due to the significant revision, it may be easier to read the untracked version. We hope that the current version is intelligible.

Reviewer #1: In Manuscript PONE-D-23-07486, Xie and colleagues present their research examining the effect of adding dexmedetomidine (DEX) with other general anesthetics, in order to create a more readily reversible anesthetic. Overall, the authors are to be commended on the amount of data presented in this manuscript, however, this large volume of data obscures issues with the design of their experiments. There are almost three separate papers and hypotheses tested. The experimental design would have been better served if the author concentrated on one general anesthetic (e.g., sevoflurane) and performed more complete experiments. Overall, the results are confusing as presented. It appears that the authors wanted to test whether the addition of DEX to a standard general anesthetic can reduce the dose of general anesthetic required. However, the authors tested four different general anesthetics using a model for anesthetic effects (loss of tail clamp response) and recovery (recovery of righting reflex – RORR). This model was then further examined using a surgical model (abdominal incision).

Due to the large amount of data, their message is confusing to discern. The paper would be better suited for publication if the paper was presented as a single general anesthetic study that was more complete. A subsequent paper could build from these findings, and further examine other general anesthetics. Specific comments are below.

The manuscript was revised to make it more transparent and easier to read. In the earlier version of the paper, we showed that supplementing Dex with propofol, isoflurane, sevoflurane or N2O all produced a potent anesthetic. In the revised study, we have taken isoflurane and N2O out of the study and they will be used in a future publication, as suggested by the reviewer. Propofol and sevoflurane, the two most popular anesthetics in the clinics, now represent the exclusive focus of the revised manuscript. We believe it important to show that both intravenous and inhalational anesthetics can be used to supplement Dex. 

The reviewer wrote that “It appears that the authors wanted to test whether the addition of DEX to a standard general anesthetic can reduce the dose of general anesthetic required.”. This comment is the key to our study since other studies have already shown that Dex could reduce the doses needed to produce anesthesia for standard general anesthetics, like sevoflurane in animals and humans. Our approach is different. Our main aim in this study was to use Dex as the primary agent to produce a target specific anesthesia. We tested high dose Dex by itself. Unfortunately, Dex alone did not completely suppress responses to noxious stimuli, nor was it completely reversible. We then assessed whether low dose propofol or sevoflurane converted a more modest dose of Dex into an anesthetic. In our study the primary agent was Dex, since the goal all along was to minimize propofol or sevoflurane. Propofol was used at doses which did not produce a loss of the righting reflex and sevoflurane at a dose that were less than ½ a MAC. At this dose of sevoflurane, all rats reacted strongly to the tail clamp stimuli (Stable 1) and some rats recovered their righting reflex. We designed the experiments to test this Dex based anesthetic strategy and found that it produced a state of general anesthesia appropriate for surgery. Finally, we tested whether the Dex based anesthetic combinations could be reversed in rats using low dose atipamezole and caffeine. It was very effective. This may potentially allow us, or others, to translate such an anesthetic strategy in the future to the human population since none of the drugs were used at doses that should elicit significant unwanted effects.

1. The authors wished to examine if a low or high dose of DEX could enhance a general anesthetic’s effects, thus decreasing the dose (MAC) of anesthetic used. The authors first determined the MAC of isoflurane and sevoflurane. Interestingly, the authors found that their MAC was higher than those published previously by others using a rat model and explain the discrepancy as being due to the tail clamp stimulus.

Tables 1A/1B should be moved to Supplemental Data. It is background information that could be moved to decrease the size of the paper.

Table 1B (isoflurane) no longer pertains to the revised manuscript and has been removed from the manuscript. Table 1A has been moved to supplemental data as requested (it has become STable 1).

The revised manuscript revisits this issue. We took the differences of MACs between our study and others in the literature seriously. We applied the tail clamp at the position of the tails and the duration consistently. We were confident of the gas concentrations since we employed a gas analyzer for every experiment. Even so, the setup was calibrated on two separate occasions. It was properly calibrated. Some of the difference may be due to the strength of noxious stimulus used between our study and the studies by others. More likely the main difference between the MAC values in the literature and those we report in the manuscript is due to the difference in the sampling sites. In our study, a nose cone was connected to the outlet port of the gas chamber by a short length of corrugated tubing. The gas concentration was sampled at the inlet to the corrugated tubing by a gas analyzer and are equivalent to measurements in a sealed gas chamber and alveolar gas concentrations, as were done by others. See the manuscript by White et al showing the difference between different kinds of measurements, in that case between an anesthesia box and alveolar gas at the trach site [1]. In those studies, the concentrations for MAC were measured in the sealed gas chamber or the end-tidal concentrations sampled at the connector to the trachea. Our measurement was only meant to confirm the EC50 or MAC equivalence of our experimental system, not to redefine the MAC for Sevoflurane. This discrepancy is now covered on lines 370-380 (untracked version). 

2. In a similar manner, the authors determined a “low dose” and “high dose” of propofol, but they did not use the tail clamp test, instead they used “consciousness.” Why did they not use the tail clamp test to be more consistent?

In the revised manuscript we explain that when Dex was supplemented with propofol the rats lost all response to the noxious tail clamp or the surgical incision at propofol doses too low to elicit unconsciousness when used by itself. If we defined these doses as “high dose propofol,” it would be confusing This issue is discussed on lines 288 and 298 (untracked version). 

3. The data is hard to read as written. For example, page 14 (Paragraph starting with “Table 2”) presents a lot of data with associated p values, and it just doesn’t read well. This issue can be found throughout the results section, and I would suggest that these sections be rewritten without so much data. Also, why did the authors study 200 and also 300 mcg/kg/min of propofol if 300 mcg/kg/min

 was considered to be low dose. In essence, two “low doses” of propofol were studied.

The information is now presented in a truncated and clearer manner. The revised manuscript explains that we were trying to identify the minimum dose of propofol that when combined with Dex, suppressed all response to a noxious stimulus. For this study, we started with 200 µg/kg/min of propofol. When that proved to be an insufficient dose, the dose was increased to 300 µg/kg/min. The latter dose was enough propofol to suppress all responses to tail clamp in both female and male rats. This is explained on lines 312 to 332 of the revised manuscript (unmarked version). That is why there are two doses of propofol. The statistics have been edited to make them clearer. Instead of describing each time point in detail, the revised manuscript only describes the aggregated data from all the time points (see the paragraph starting at line 312). Nonetheless, Tables 2 and 3 still contain all the statistics for each time point for any interested reader.

 4. The authors also wanted to see if DEX could be reversed using a commonly known and utilized antagonist, atipamezole. This drug has been used extensively in the veterinary community for reversal of DEX or its non-racemic mixture medetomidine. Thus I am not sure why the authors present atipamezole as a new agent. It may not work as well in humans (as they mention in their discussion), but it is not novel in my opinion.

If the authors wish to study reversal of DEX effects, it should not have been included in the same study as examining “low dose” propofol. This idea should have been tested separately.

We recently published a study showing that atipamezole and caffeine when used together, rapidly and completely reversed Dex sedation [2]. In that study, we reported lower doses of Atipamezole (1/10 or 1/20 of the manufacturer’s recommended dose) could reverse Dex’s sedative effect rapidly when it was combined with caffeine in rats. If that strategy can be translated to humans, then low dose Atipamezole supplemented with caffeine has the potential to overcome the high dose requirement of Atipamezole thereby minimizing, or even eliminating, the unwanted effects of high dose Atipamezole. We do not have any human data for low dose Atipamezole with caffeine. Even so, we believe that the concept of using low dose Atipamezole with caffeine to be promising. In the current study we supplemented Dex with a low dose of a second agent to create a potent anesthetic, which produced immobility, antinociception, and potentially interfered with memory. Without a reversal agent for these Dex based combinations, the utility in humans would be in doubt. We evaluated atipamezole and caffeine to ensure that it was still capable of reversing Dex with the second agent. It worked perfectly. Dex supplemented with a low dose of a second agent represents a rapidly reversible potent anesthetic. To us that is part of the same story; a drug combination that may be safer than currently employed anesthetics, and which is reversible within seconds. We believe that this is worth reporting the Dex based anesthesia and its reversal together.

Atipamezole was originally evaluated for use in the human population over 30 years ago. For 30 years it has not been employed in humans due to its unfortunate side effect profile at the high doses required to reverse Dex. By adding caffeine, dosages 20-fold lower of atipamezole effectively reversed Dex compared to the doses needed when it is used by itself.

For 30 years no one used atipamezole in humans; the drug was a failure in human medicine. We do not claim that atipamezole is a novel reversal agent. We show a novel strategy for using it that should work in humans as well as rodents. Without effective reversal, expanding the reach of Dex may be problematic. Emergence times are just too slow. That is why we believe that it is important that the reversal data remain in the manuscript. See lines 471-481 of the discussion.

5. The paper would have been better organized by studying one general anesthetic completely, without reversal. Then, they could have added a section examining DEX reversal from a general anesthetic. Similarly, the abdominal surgery portion of the study, could have been done with a single general anesthetic agent, not four general anesthetics.

As noted above in the comments to the editor, we removed two anesthetics (isoflurane and N20) to make the paper more accessible, as requested by the reviewer. While we showed the abdominal surgeries were successfully performed under Dex with low propofol. We do not yet know which combination will prove to be the “safest.” That will require additional testing. See line 587 in the Discussion.

6. Low and high dose DEX data (Figure 10 and Table 5) should be presented first, as the showing this effect should have been the initial study done before examining combinations of DEX with general anesthetics.

Thank you for the suggestion. It made the flow better. The Results section of the manuscript starts with the comparison between low and high dose Dex, as requested. See the beginning of the Results section starting with line 245.

7. The data on using DEX combined with propofol, isoflurane, and sevoflurane for abdominal surgery does follow their prior data using a tail clamp, both are painful stimuli, but is the lack of change in vital signs proof of a complete anesthetic? This data could be removed without affecting their interpretation.

We had the same discussion amongst ourselves. We thought that working anesthesiologists would not necessarily appreciate the pain involved in a noxious tail clamp. They would only believe that we observed immobility if and only if it took place during a painful surgery, as that type of procedure remains the gold standard for a successful anesthetic. Abdominal surgery represents such a painful surgery. We agree with the reviewer that lack of change in vital signs alone is not the proof of a complete anesthetic. While the animals were unconscious, the lack of vital sign change during tail clamp or surgery may represent the antinociceptive property of anesthetics. We discuss why surgery is part of the revised manuscript on line 411 of the Results section. As the reviewer suggested, we only show one anesthetic combination for the surgery in the figure to reduce the size of the MS.

8. The EEG analysis is quite complex, and it appears that it is included to demonstrate that the DEX/general anesthetic combinations effect memory and awareness. Again, this data could be a separate paper and by combining it with their other data, the resulting message/interpretation is confused. Why did they feel the need to demonstrate these effects?

Again, the combination of Tail clamp experiments with four different general anesthetics, abdominal surgery, and EEG analyses seems unnecessary to demonstrate that DEX can enhance lower dose general anesthetics to produce an “anesthetized” state.

The isoflurane and N2O data were removed, which shortened the manuscript and reduced the number of figures. In the past, our studies were criticized because they did not contain EEG data which precluded exploration of certain kinds of mechanism. We are sure that if we remove the EEG data, that EEG researchers will critique the study. The predominant presence of slow wave (delta, 0.5-4 Hz) is a consistent feature in the surgical state of general anesthesia [3] . Without the ability to directly measure amnesia and awareness in rats, we used the EEG recorded at 1 MAC of sevoflurane or isoflurane as a reference, since these volatile agents ~1 MAC are thought to reach EC100 for amnesia and awareness. The EEG data allowed us to confirm the power of the slow wave in Dex based anesthesia and allowed a comparison of the EEGs between the volatile agents at 1 MAC and the Dex based anesthetic. The persistent presence of the slow wave in EEG further confirmed these animals were in the stage of general anesthesia when they were exposed to the Dex based anesthetic. More importantly, our EEG data suggests that memory may be impaired when Dex is supplemented with either propofol or sevoflurane, a requirement for a general anesthetic. However, the final confirmation of memory under these Dex combinations can only be determined in humans. Because it is in a separate section of the Results section, we feel we should provide the basic pattern of EEG for readers who are interested in the EEG information. Anyone else can simply skip over this section.

9. The protocols used seem a bit random. For example, rats were anesthetized with isoflurane prior to surgery required for measuring EEG, then DEX was given, and the animals given propofol boluses and then an infusion. Various drugs were given after 30 or 60 minutes (why these timepoints?), infusion doses were changed (e.g., DEX was dropped from 15 mcg/kg/hr to 12 mcg/kg/hr) but why?

In the revised Methods, the rationale for the protocol employed is presented in more detail (see line 167). For example, we used isoflurane simply to render rats’ unconscious such that they could be weighed and an I-V line inserted. In some experiments the EEG electrodes were inserted, and the EEG was recorded at ~ 1 MAC for the comparison under isoflurane. We waited for the isoflurane to wash out before starting our studies of tail clamp and vital signs. In the sevoflurane experiments we used sevoflurane throughout, but it is 10X more expensive than isoflurane, so it was not used in all experiments. 

The revised manuscript has been normalized in terms of drug dosages. In the revised manuscript there are no drugs given at 30 minutes. Only at 60 minutes. 

In our response to comment #3, above, we explained that we tried to optimize Dex and propofol to achieve the best possible effect. We had two different goals. One was to completely suppress any response to a noxious stimulus, while the second was to minimize the Atipamezole levels needed for complete and rapid reversal of anesthesia. We could achieve EC100 in preventing response to tail clamp stimuli by using Dex at 12 mcg/kg/hr with propofol at 300 mcg/kg/min. We used these dosages for the rest of the study, which also allowed us to lower Atipamezole levels. 

10. The Discussion describes a lot of prior work on DEX reversal agents, which doesn’t appear to be the major point of the manuscript and has been previously published by this laboratory (Ref. 42). Again, this focus on DEX reversal takes away from their findings.

In our previous study, we focused mainly on the effect of low Atipamezole with caffeine to reverse the sedative effect of a single dose of Dex and a protocol mimicking Dex sedation for MRI. In this study, we aimed to create an anesthetic strategy which is more target specific and reversible. Emergence from Dex sedation can be slow. Will the recovery of Dex based anesthesia be even slower but still reversible? For example, the rats took ~1 hour to “wake” from Dex supplemented with propofol. And then they were sluggish for an extended duration. Without active reversal it is possible that no one will want to use a Dex based anesthetic even if it is safer. The fact that supplemented Dex can be reversed is an important part of the anesthetic story. We have tried to make this point clearer in the revised Discussion (see line 574). The previous study showed that atipamezole and caffeine reversed Dex by itself. Without the new data contained in this manuscript, everyone would be left wondering whether atipamezole and caffeine also reversed the general anesthetic effects of Dex with propofol or Dex with sevoflurane. 

11. The authors state that they wish to examine if combining DEX with general anesthetics can allow for “sub-therapeutic” doses of the general anesthetic. In fact, other published work has already demonstrated that DEX can reduce the MAC of volatile agents in animals and humans. If this fact is known, then what does this study add to the literature? They state they their desire is to determine if DEX can be used as the primary anesthetic, but when combining it with other agents, which one is primary and which one is secondary? Or does it matter? I feel that the authors are arguing a fine point that is not as important.

The question is can one reduce the amount of general anesthetic needed (whether it is isoflurane, sevoflurane, propofol, or nitrous oxide) when combined with DEX. Again, this fact has been demonstrated so what does this study add besides a more thorough examination of multiple drug combinations?

Thank you for this comment. We did a poor job explaining the novelty of the study. It is important to us that our manuscript stand on its own. We need to get this explanation right.

Volatile anesthetics and propofol have been linked to developmental abnormalities in neonatal animals, and with cognitive issues in elderly patients. It is not yet known if the neonatal studies in animals are relevant to the human population. Even with that caveat, it is likely that if safe and effective general anesthetics were available that did not use substantial amounts of sevoflurane, isoflurane, propofol or ketamine, they would supplant these popular general anesthetics. Dex is a safer agent which is why it is common in pediatric sedation.

The studies using Dex to lower the MAC concentration for sevoflurane are undoubtedly trying to reduce the sevoflurane concentration used in surgeries thereby creating safer anesthetics. Our goals were different, even though it is also about creating a safer anesthetic. We were hoping to eliminate the need for the current generation of anesthetics by employing a high concentration of Dex by itself. Does high dose Dex represent a suitable general anesthetic if it is reversible? Would the animals exhibit extreme bradycardia or hypotension? Could high dose Dex be successfully reversed with atipamezole & caffeine? Unfortunately, these studies did not work out. The good news is that high dose Dex did not produce any more bradycardia or hypotension than did lower doses. Unfortunately, some rats still responded to the tail clamp. And although atipamezole & caffeine re-established the righting reflex quickly, within a minute, the animals were sluggish and barely moved for a long time. Reversal was therefore incomplete. In clinical practice, a higher total dose of Dex means a higher dose of Atipamezole for reversal. Higher doses of Atipamezole are not our intent. 

Our next goal was to determine the absolute minimum levels of a second agent (isoflurane, sevoflurane, propofol or N2O) would convert a more modest dose of Dex to an effective anesthetic. One that could then be effectively reversed. We were successful in that goal. The second agents were employed at sub hypnotic levels. See line 508 of the discussion.

We explain in the Discussion the relevance of our study relative to what was done previously. In those earlier studies, the primary agent was sevoflurane. In our study the primary agent was Dex. Only a small amount of sevoflurane or propofol was employed. The neuroapoptosis triggered by general anesthetics should be dose dependent. Using a sub hypnotic dose should be as safe as possible. Especially since we tried to identify the lowest effective dose possible. 

Finally, we observed that supplementing Dex with any anesthetic produced an effective anesthetic. We evaluated propofol and sevoflurane (but also isoflurane, and N2O). We made no assumptions about which of these combinations was the safest. Our goal, in future studies, is to determine which of these combinations is the safest. It may not necessarily be Dex with sevoflurane.

Please see Discussion line 581.

12. The authors discuss a major limitation of their study, simply put, they only studied female rats. It has been shown that female rats are more sensitive to DEX effects than male rats. This issue seriously limits the interpretation of their data.

What the authors do not discuss completely in my opinion is why their data demonstrates such a major effect of DEX compared to prior work.

We performed additional experiments in male rats for this revision. Data from male rats is now included in the revised manuscript. In our earlier study we saw no clear differences between male and female rat responses to Dex’s sedative effect, although there may have been a trend in the direction of males being less sensitive and we had simply not tested a large enough cohort of rats [2]. In the current study we saw no difference between male and female rats when testing combinations of drugs (see Table 3 which compares data from Male and Female rats and Supplementary Figs 3 & 4). Data from male rats is now covered throughout the manuscript. Examples are found on lines 392 and 398.

Reviewer #2: This is a feasibility study in rodents to test whether combining dexmedetomidine with low doses of conventional anesthetics is sufficient to provide surgical anesthesia. The rationale for the study is that conventional anesthetics are known to cause delirium and cognitive dysfunction in elderly patients, and dexmedetomidine is known to be less deleterious. The manuscript is well written, and the results are described clearly. However, I have several concerns that need to be addressed.

• The short title, “A strategy for creating a new anesthetic,” is misleading. “New anesthetic” implies a novel drug, but the authors describe a novel dosing regimen using existing anesthetics. This should be revised.

The revised manuscript has a short new title. “A novel dosing regimen creates an effective dexmedetomidine based anesthetic.” 

• As stated in the Abstract and Introduction, the premise of the study is that dex administration will allow for lower doses of conventional anesthetics that are associated with delirium and cognitive dysfunction in the elderly. However, the authors did not use aged animals and did not test for delirium or cognitive dysfunction in their study.

The reviewer is correct. Aged animals would be preferable. Nonetheless, working with aged animals is difficult and expensive. Our goal was to find out whether this strategy would work at all before going to the time and expense of studying older animals. In future studies we hope to reproduce these studies in aged animals of both sexes. This is discussed on line 589.

• A significant portion of the Introduction discusses the methods, results, and conclusions of the study. This content belongs in the Methods, Results, and Discussion sections, respectively. The Introduction should focus on the background and rationale.

The revised Introduction focuses on the background and rationale for the studies. Some results are included as well.

• Why were only female rats used for the study? The NIH and most journals now require the use of both sexes to account for sex as a biological variable.

The revised manuscript includes data from male rats. As you can see from Table 3 and SFigs 3 & 4, there was no clear difference between the sexes. 

• Were the anesthetic exposures conducted in random order?

Drugs were applied in a randomized manner. This is found in the methods on line 245.

• It should be clearly stated in the manuscript that atipamezole is not approved for human use. This greatly limits the translational potential of these results to the clinical setting.

The original manuscript contained that information. So does the revised manuscript, paragraph starting on line 471. Nevertheless, this does not limit the translational potential. Quite the contrary. In previous human trials atipamezole was evaluated at a variety of dosages. At the high dosages required to reverse Dex, it produced significant side effects. Too many for regulatory approval. It was also evaluated at lower dosages. At these dosages (<30 mg) it produced no unwanted effects, but it was not able to reverse Dex. In our studies we are using even lower dosages (0.5-1: 1 ratio) than those that produced no unwanted effects. Nonetheless, the drug effectively reverses Dex. Why? It is the caffeine which increases [cAMP]i countering the Dex’s effect to lower [cAMP]i (Ref 1 in this letter). When used together extremely low doses of atipamezole become effective. We believe that this combination has strong translational potential.

• There are far too many figures. Many of them should be combined.

Some data from the original paper has been removed. Some data has moved to Supplementary material. Several Tables have been combined, as requested. For example, see Table 3 which combines most of the tail clamp data. 

• I find it curious that MAC values for sevoflurane and isoflurane were much higher than reported values in the literature. The authors attribute this to their tail clamp being a more potent noxious stimulus, but are other explanations possible? Were the vaporizers and agent analyzers properly calibrated? Maybe the equilibration times were too short? 

The revised manuscript revisits this issue (please also see the response to the first reviewer). We too worried about the differences between our study and values in the literature. Although we were confident of the gas concentrations, since we employ a gas analyzer for every experiment, we still had the setup calibrated on two separate occasions. It was properly calibrated. Some of the difference may be due to the powerful noxious stimulus used. More likely most of the difference between the MAC values in the literature and those we report in the manuscript are due to measurement differences. In our study, a nose cone was connected to the outlet port of the gas chamber by a short length of corrugated tubing. The gas concentration was sampled at the inlet to the corrugated tubing by a gas analyzer and are alveolar gas concentrations, as was done by others. See the manuscript by White et al showing the difference between different kinds of measurements, in that case between an anesthesia box and alveolar gas sampled at the tracheostomy [1]. Our measurement was only meant to confirm the EC50 or MAC equivalence of our experimental system, not to redefine the MAC for Sevoflurane. This discrepancy is now covered on lines 370-380 (untracked version). 

• In the Discussion, it seems arbitrary to call dex the “primary” anesthetic agent in these studies. The study showed that combining sub-anesthetic doses of dex and conventional anesthetics is sufficient to produce surgical anesthesia, so in my view, neither is the “primary” anesthetic.

This is more than a semantic argument on our part. When used by itself, Dex at high dosages is not enough to produce anesthesia in every animal tested but is enough to produce a deep level of unconsciousness. The only thing that will rouse the rats is a powerful noxious stimulus. Otherwise, the rats are “out.” They do not respond to sound or touch. The levels of sevoflurane or propofol used to supplement the Dex, were not, by themselves, sufficient to elicit unconsciousness or if it did so, any stimulus, such as IV insertion, would rouse the animals. 

Our goal was to use a large dose of Dex so that we would not have to use much of the second agent. If the second agents are toxic, then minimizing their dosage represents an important objective. This point is covered in the discussion starting on lines 503-510.

References:

1. White PF, Johnston RR, Eger EI. Determination of Anesthetic Requirement in Rats. Anesthesiology. 1974;40: 52–57. doi:10.1097/00000542-197401000-00012

2. Xie Z, Fox AP. Rapid emergence from dexmedetomidine sedation in Sprague Dawley rats by repurposing an α2-adrenergic receptor competitive antagonist in combination with caffeine. BMC Anesthesiol. 2023;23: 39. doi:10.1186/s12871-023-01986-5

3. Akeju O, Brown EN. Neural oscillations demonstrate that general anesthesia and sedative states are neurophysiologically distinct from sleep. Curr Opin Neurobiol. 2017;44: 178–185. doi:10.1016/j.conb.2017.04.011

---

## [Decision Letter · Decision Letter 1]

13 Aug 2023

PONE-D-23-07486R1Towards A Potent and Rapidly Reversible Dexmedetomidine-Based General AnestheticPLOS ONE

Dear Dr. Xie,

Thank you for submitting your manuscript to PLOS ONE. After careful consideration, we feel that it has merit but does not fully meet PLOS ONE’s publication criteria as it currently stands. Therefore, we invite you to submit a revised version of the manuscript that addresses the points raised during the review process.

ACADEMIC EDITOR:please carefully assess all the reviewers comments

We look forward to receiving your revised manuscript.

Kind regards,

Silvia Fiorelli

Academic Editor

PLOS ONE

Reviewers' comments:

Reviewer's Responses to Questions

**Comments to the Author**

1. If the authors have adequately addressed your comments raised in a previous round of review and you feel that this manuscript is now acceptable for publication, you may indicate that here to bypass the “Comments to the Author” section, enter your conflict of interest statement in the “Confidential to Editor” section, and submit your "Accept" recommendation.

Reviewer #1: (No Response)

Reviewer #2: All comments have been addressed

2. Is the manuscript technically sound, and do the data support the conclusions?

Reviewer #1: Yes

Reviewer #2: Yes

3. Has the statistical analysis been performed appropriately and rigorously? 

Reviewer #1: I Don't Know

Reviewer #2: Yes

4. Have the authors made all data underlying the findings in their manuscript fully available?

Reviewer #1: Yes

Reviewer #2: Yes

5. Is the manuscript presented in an intelligible fashion and written in standard English?

Reviewer #1: Yes

Reviewer #2: Yes

6. Review Comments to the Author

Reviewer #1: In Manuscript PONE-D-23-07486_R1, Xie and colleagues have revised their prior manuscript as per the reviewers’ comments. Overall, I appreciate the amount of work and new experiments performed for this revision. By concentrating only one two general anesthetic agents, sevoflurane (sevo) and propofol (prop), the authors have made the paper more focused and thus their conclusions are more easily appreciated. However, there are still some issues that need to be addressed (without more experiments) as discussed below.

1. The introduction is much improved in flow and focus. It is now clear that the authors are attempting to extend prior research on this topic. Previously, others have shown that co-administration of dexmedetomidine (dex) can reduce the MAC (or concentration) of sevo or propofol necessary to achieve an anesthetic state. Namely, the authors are attempting to discern if higher doses of dex co-administered with sevo, or prop can achieve an anesthetic state while using a sub-anesthetic dose of the same.

By describing why high dose dex alone as an anesthetic agent has failed in the past (e.g., slow emergence, lack of good reversal agents, vital sign effects), it becomes more apparent why further work is needed. However, I still struggle with the idea that the combination of dex with other agents can lead to dex being the “primary” agent. For example, prior work has shown that dex can reduce the MAC of another agent. In this work, the authors have also shown that dex can reduce the MAC of another agent, possibly reducing the second agent (sevo or prop) to a concentration/dose that is now considered sub-anesthetic. In other words, the authors appear to not be doing something unique but rather extend prior work to see if the MAC lowering effect of dex can be extended to reduce MAC further by using a higher dose of dex than that studied previously.

2. The data reported in this study strongly suggests that the effect of dex to reduce MAC can be seen at higher doses of dex. The study then is extended to demonstrate that the effects of dex/sevo or dex/prop in a rat model of tail clamping (used to simulate surgical stimulation) can be extended to a sham surgery protocol. This protocol was included to assess if the behaviors seen in the rat model are similar to a surgical protocol. As noted by the authors, they could only measure changes in vital signs as a surrogate for anesthetic effects, but the data appears to support their hypothesis

3. One drawback to the use of dex at higher doses is the need for a reversal agent. As there is no reversal agent approved for humans use, the authors attempted to reverse dex with atipamezole and caffeine (as done in their prior published work). It is clear that this combination of reversal agents works.

4. Lastly, in order to determine if the newer combinations of dex with either sevo or propofol produce amnesia, the authors opted to study these anesthetic combinations with EEG analysis. There is a lot of data and I appreciate that the authors feel it is important to include this data due to concerns that potential reviewers may not agree with their findings without evidence of amnesia.

5. Therefore, the current revision is much improved and more focused in design, but not more focused in the writing. The paper contains large sections of discussions of early data. For example, Page 90 Paragraph 2 is very long to simply make the point that dex suppressed responses to noxious stimuli. These discussions and tables (Table 1-3) are not necessary or should be moved to the supplemental data area. The paper is still too long for the conclusions reached and such detail makes reading the paper difficult.

6. If the authors feel the need to retain the section of EEG analysis, it needs to be shortened. Does all of the detail need to be included?

7. On Page 111 Lines 706-717, the authors concisely summarize their work and results. However, this is buried in the discussion. Similarly, the section on EEG effects (Page 112 Lines 721-738 is summarized nicely. This style of writing is appreciated for its ease of readability. The authors should look for other sections that could similarly be reduced.

8. The figure legends often contain extremely long descriptions of the statistics derived from their work. It isn’t necessary and is impossible to read. If the authors feel this data is needed, then it should be in a supplemental figure/table or somehow included in the figure itself. For example, Figure 5 on Page 121 Line 928-961 is impossible to read. Other figure legends are similar.

9. Overall, a great revision but still central message is lost by the length and unnecessary detail included in the manuscript. The story is now clearer but the writing isn’t.

a) Is it possible to use dex as a primary agent (i.e., a higher dose)?

b) Can we overcome limitations of using higher dex doses (no reversal, slow emergence)?

c) Can co-administration of higher doses of dex further lower the amounts of other general anesthetic agents utilized (e.g., sevo or prop) as suggested by prior work?

d) If so, do such combinations of high dex with sevo or prop produce the other aspects of a general anesthetic (i.e., ability to perform surgery, effects on memory/amnesia)?

Reviewer #2: (No Response)

7. PLOS authors have the option to publish the peer review history of their article (what does this mean?). If published, this will include your full peer review and any attached files.

Reviewer #1: **Yes: **Timothy Angelotti MD PhD

Reviewer #2: No

---

## [Author Response · Author response to Decision Letter 1]

18 Aug 2023

PONE-D-23-07486R2

Towards A Potent and Rapidly Reversible Dexmedetomidine-Based General Anesthetic

August 18, 2023

Dear Dr Fiorelli,

We would like to thank the reviewers for their diligence. Multiple changes were made in response to the reviews to address the comments. Their comments will definitely help to improve our MS. 

This letter includes a point-by-point response to each of the comments made by the reviewers and the editor. The reviewer’s comments are in bold italics followed immediately by our reply in regular font. The line numbers in the reply to the reviewers correspond to the line numbers in the version with “Tracked Changes – Simple Markup.”

Reviewer #1: In Manuscript PONE-D-23-07486_R1, Xie and colleagues have revised their prior manuscript as per the reviewers’ comments. Overall, I appreciate the amount of work and new experiments performed for this revision. By concentrating only one two general anesthetic agents, sevoflurane (sevo) and propofol (prop), the authors have made the paper more focused and thus their conclusions are more easily appreciated. However, there are still some issues that need to be addressed (without more experiments) as discussed below.

1. The introduction is much improved in flow and focus. It is now clear that the authors are attempting to extend prior research on this topic. Previously, others have shown that co-administration of dexmedetomidine (dex) can reduce the MAC (or concentration) of sevo or propofol necessary to achieve an anesthetic state. Namely, the authors are attempting to discern if higher doses of dex co-administered with sevo, or prop can achieve an anesthetic state while using a sub-anesthetic dose of the same.

By describing why high dose dex alone as an anesthetic agent has failed in the past (e.g., slow emergence, lack of good reversal agents, vital sign effects), it becomes more apparent why further work is needed. However, I still struggle with the idea that the combination of dex with other agents can lead to dex being the “primary” agent. For example, prior work has shown that dex can reduce the MAC of another agent. In this work, the authors have also shown that dex can reduce the MAC of another agent, possibly reducing the second agent (sevo or prop) to a concentration/dose that is now considered sub-anesthetic. In other words, the authors appear to not be doing something unique but rather extend prior work to see if the MAC lowering effect of dex can be extended to reduce MAC further by using a higher dose of dex than that studied previously.

Thank you for your thoughtful comments. The reviewer correctly pointed out that “The authors are attempting to discern if higher doses of Dex co-administered with sevo, or prop can achieve an anesthetic state while using a sub-anesthetic dose of the same.” The reviewer is also correct that “prior work has shown that dex can reduce the MAC of another agent.” Most of the previous studies using Dex to reduce MAC levels were done in humans (Ref 61,63,64). Here is a summary of the previous studies in question: Dex was used an adjunct agent at the doses ranging from 0.3-0.7 mcg/kg/hr which reduced the MAC concentration of sevoflurane. In those studies, the modest dose of Dex that was employed, reduced the MAC of sevoflurane by 20-30%. In one pediatric study, low dose Dex (0.2 mcg/kg/hr), reduced the required sevoflurane dose by 60%. However, these patients also received a caudal block after induction for the procedure, thus obviating the need to provide analgesia (Ref 62). In another study with isoflurane, higher doses of Dex (up to 2.85 mcg/kg/hr) were used in human volunteers. The MAC of isoflurane was reduced by 50%. However, the recovery time was very long (up to 4-6 hours) in those subjects (1, see reference below; We did not cite this article because we focused on sevoflurane and propofol in the MS). In a case report with three patients, higher doses of Dex (5-10 mcg/kg/hr) alone were used in conjunction with local anesthetics (ref 60) for surgery. Dex at these high doses, which were like the Dex dose (12 mcg/kg/hr) used in rats in our study, produced sufficient anesthesia for the procedures without significant hemodynamic compromise in the three patients. Again, their recovery time was about 2 hrs or longer. 

Studies of high dose Dex have been hindered in the adult human population for two reasons. First, when Dex is used in the adult population the dosages are typically low. It is employed at dosages in the range 0.2 - 0.7 mcg/kg/hr. These low dosages already produce significant hemodynamic effects. The worry is that going to higher dosages will exacerbate the hemodynamic effects and this concern prevents higher doses being employed in adults. Second, as you can already see from the brief review of the literature that was presented above, higher doses of Dex produce enormously long emergence times in humans. Hours, to many hours is simply not acceptable. No one will use high dose Dex, even if it is demonstrably safer. It’s just not practical.

In rats, our studies show that the hemodynamic effects of Dex saturate at low concentrations. High dose Dex does not produce additional hemodynamic changes compared to low dose Dex. Most importantly, using our reversal agent, we change the emergence times from hours to seconds. Our results suggest that high dose Dex can be used as the primary agent safely and without tying up recovery facilities for hours. We believe that these results may change the way that clinicians look at Dex.

We agree with the reviewer that on one level we could interpret our study as an extension of some of the above previous work by using a higher dose of Dex to permit a further reduction of the required doses of sevoflurane or propofol. However, we would argue instead that our focus driving this work was not so much the reduction of the required quantities of other common anesthetizing agents using Dex as an “adjunct”, but rather the use of minimal amounts of other agents to allow Dex to function as an effective general anesthetic. On the surface, this characterization appears semantically tautological- much like the distinction between characterizing a glass as “half empty” vs “half full”. We would like to appeal to the very different philosophical and perspectival orientations that underlie the use of one description vs the other. Without an effective reversal for Dex, the use of higher doses of Dex will not be feasible clinically. The use of Dex will be limited at 0.2-0.7 mcg/kg/hr which will not produce unconsciousness in humans. It will be only used as the less potent agent of two agents regardless the combinations. In pediatric sedation, Dex is commonly used at 2-3 mcg/kg/hr. At this dose, most patients are deeply sedated or unconscious. At these doses of Dex, high doses of sevo or propofol are not needed. However, this high dose of Dex prolongs recovery if there is no safe and effective reversal. The unique nature of our study is that we can create a reversible anesthetic combination with more selective targets. We believe the higher the dose of sevoflurane or propofol the less selective they are. In this study, we cannot confirm whether higher dose of Dex and lower dose of sevo or propofol are safer in the elderly or less neurotoxic in neonates. We only confirm this approach is feasible and we will assess these combinations to determine their safety, efficiency and neurotoxic profiles in humans and animals in the future.

The reviewer pointed out that any agent could in principle be considered “primary” in our combinations because we require both Dex and the second agent to maintain robust general anesthesia. To justify our characterization, consider the analogy to how one assigns which component in a liquid/liquid solution is the “solvent” (since one could argue that either liquid could be considered a solvent). Conventionally this assignment is made based on the component that is present in greater chemical quantity. Similarly, we call Dex the “primary” agent in our anesthetic combinations because the Dex was present in greater “anesthetizing quantity” than the other agents with which it was paired. By this we mean that while Dex alone at the doses we used was sufficient to keep all rats unconscious for the duration of infusion, the second agents (propofol or sevoflurane) at the doses we used were unable by themselves to maintain unconsciousness and suppress righting in the rats. In addition, Dex was the drug we reversed. The doses of sevo or propofol were so low that no reversal for sevo or propofol was needed in these combinations.

As we extend our findings into human subjects, we will seek to establish an effective and safe dose in humans which is comparable to the doses of Dex in this animal study. We might start, for example, at 0.7-1 mcg/kg/hr. Pediatric sedation has employed doses up to 3 mc/kg/hr safely. Very likely the sweet spot will be somewhere in between. Armed with an effective reversal, if it proves to be effective, we will be empowered to evaluate Dex at doses higher than those we currently use when employing Dex as an adjunctive agent. 

Finally, we defined “low dose” dex as 10 mcg/kg bolus and 12 mcg/kg/hr infusion vs “high dose” as 40 mcg/kg and 48 mcg/kg/hr based on the responses to tail clamp stimuli engendered. Dex at 10 mcg/kg and 12 mcg/kg/hr is considered a very high dose in human despite a case report (Ref 62) documenting comparable doses. In rat studies this dose is on the low end. With this low dose, all rats lost their righting reflex and remained unconscious, but not be able to tolerate any noxious stimuli, including IV insertion. In the study of amphetamine and Dex (ref. 77), those authors used a 50 mcg/kg bolus. It is used at even higher doses in veterinary medicine. We attempted to use the lowest dose of Dex which caused a loss of righting reflex to combine with other subanesthetic doses of sevo or propofol to produce reversable general anesthesia. 

We have included these points in the discussion section (please see lines 504-531).

2. The data reported in this study strongly suggests that the effect of dex to reduce MAC can be seen at higher doses of dex. The study then is extended to demonstrate that the effects of dex/sevo or dex/prop in a rat model of tail clamping (used to simulate surgical stimulation) can be extended to a sham surgery protocol. This protocol was included to assess if the behaviors seen in the rat model are similar to a surgical protocol. As noted by the authors, they could only measure changes in vital signs as a surrogate for anesthetic effects, but the data appears to support their hypothesis.

Thank you for your understanding.

3. One drawback to the use of dex at higher doses is the need for a reversal agent. As there is no reversal agent approved for humans use, the authors attempted to reverse dex with atipamezole and caffeine (as done in their prior published work). It is clear that this combination of reversal agents works.

Thank you very much. 

4. Lastly, in order to determine if the newer combinations of dex with either sevo or propofol produce amnesia, the authors opted to study these anesthetic combinations with EEG analysis. There is a lot of data and I appreciate that the authors feel it is important to include this data due to concerns that potential reviewers may not agree with their findings without evidence of amnesia.

Thank you for your understanding. 

5. Therefore, the current revision is much improved and more focused in design, but not more focused in the writing. The paper contains large sections of discussions of early data. For example, Page 90 Paragraph 2 is very long to simply make the point that dex suppressed responses to noxious stimuli. These discussions and tables (Table 1-3) are not necessary or should be moved to the supplemental data area. The paper is still too long for the conclusions reached and such detail makes reading the paper difficult.

We revised the discussions as instructed by the reviewer. The paragraph 2 in page 90 (previous tracked version) was shortened. Tables 1-3 were moved to supplemental data area. We only described the tail clamp information in the result section. 

6. If the authors feel the need to retain the section of EEG analysis, it needs to be shortened. Does all of the detail need to be included?

We feel the EEG analysis is critical in establishing some of the conclusions of our manuscript. There is no other way to monitor brain activity and provide a compelling proxy for amnesia/awareness in the context of an ablated righting reflex. With our basic and non-invasive recording, we tried to compare EEGs between the Dex based combinations and the conventional anesthetic agents. We stated our observation without making any conclusive statement on amnesia. We feel some readers may be interested in the EEG information for the Dex based anesthesia. Some evidence suggested EEG monitoring is helpful to prevent overdose of anesthetics in certain vulnerable populations (2, 3, references below). A recent review paper suggested that intraoperative EGG monitoring is helpful to enhance recovery although there is insufficient outcome data (see reference 4 below). In our study, we compare the EEG spectrograms and power spectra with ~ 1 MAC of volatile agent vs Dex combinations and shown that the Burst-Suppression Ratio under ~ 1 MAC of volatile agent vs Dex combinations were very different.

We have shortened the EEG discussion by moving some statistical information to the figure legends. As the reviewer correctly points out, EEG data is complicated. As such we feel compelled to ensure we have provided sufficient information to the readers in the methods and analysis. We ask for the reviewer’s understanding. 

7. On Page 111 Lines 706-717, the authors concisely summarize their work and results. However, this is buried in the discussion. Similarly, the section on EEG effects (Page 112 Lines 721-738 is summarized nicely. This style of writing is appreciated for its ease of readability. The authors should look for other sections that could similarly be reduced.

Thank you for your suggestion. We revised the discussion section accordingly. 

8. The figure legends often contain extremely long descriptions of the statistics derived from their work. It isn’t necessary and is impossible to read. If the authors feel this data is needed, then it should be in a supplemental figure/table or somehow included in the figure itself. For example, Figure 5 on Page 121 Line 928-961 is impossible to read. Other figure legends are similar.

Thank you for the suggestion. We agree the extensive statistical description is not necessary and may cause confusion. We have re-written the figure 3, 5 and 7 legends and made them shorter. We used the number at the point before the application of Dex or Dex based combination as the baseline. We compared the rest of the time points to the baseline. We hope it is easier to read. 

9. Overall, a great revision but still central message is lost by the length and unnecessary detail included in the manuscript. The story is now clearer but the writing isn’t.

a) Is it possible to use dex as a primary agent (i.e., a higher dose)?

b) Can we overcome limitations of using higher dex doses (no reversal, slow emergence)?

c) Can co-administration of higher doses of dex further lower the amounts of other general anesthetic agents utilized (e.g., sevo or prop) as suggested by prior work?

d) If so, do such combinations of high dex with sevo or prop produce the other aspects of a general anesthetic (i.e., ability to perform surgery, effects on memory/amnesia)?

We appreciate these suggestions and include all of them in the discussion. 

Reviewer #2: (No Response)

Thank you for reviewing our MS.

References:

1. Khan ZP, Munday IT, Jones RM, Thornton C, Mant TG, Amin D. Effects of dexmedetomidine on isoflurane requirements in healthy volunteers.1: Pharmacodynamic and pharmacokinetic interactions Br J Anaesth. 1999; 83:372–80

2. Koch S et al. Perioperative electroencephalogram spectral dynamics related to postoperative delirium in older patients, Anesthesia & Analg. 2021

3. Yuan I et al. Prevalence of isoelectric electroencephalography events in infants and young children undergoing general anesthesia, Anesthesia & Analg. 2021

4. Chan, MTV et al. American Society for Enhanced Recovery and Perioperative Quality Initiative Joint Consensus Statement on the Role of Neuromonitoring in Perioperative Outcomes: Electroencephalography, Anesthesia & Analg. 2020

Respectful

Zheng (Jimmy) Xie, MD, Ph.D, FASA

Professor

Department of Anesthesia and Critical Care

University of Chicago

---

## [Decision Letter · Decision Letter 2]

6 Sep 2023

Towards A Potent and Rapidly Reversible Dexmedetomidine-Based General Anesthetic

PONE-D-23-07486R2

Dear Dr. Xie,

We’re pleased to inform you that your manuscript has been judged scientifically suitable for publication and will be formally accepted for publication once it meets all outstanding technical requirements.

Kind regards,

Silvia Fiorelli

Academic Editor

PLOS ONE

Additional Editor Comments (optional):

Congratulations to the authors and thanks to the reviewers for the suggestions provided which really helped improve the quality of the manuscript

Reviewers' comments:

Reviewer's Responses to Questions

**Comments to the Author**

1. If the authors have adequately addressed your comments raised in a previous round of review and you feel that this manuscript is now acceptable for publication, you may indicate that here to bypass the “Comments to the Author” section, enter your conflict of interest statement in the “Confidential to Editor” section, and submit your "Accept" recommendation.

Reviewer #1: All comments have been addressed

Reviewer #2: All comments have been addressed

2. Is the manuscript technically sound, and do the data support the conclusions?

Reviewer #1: Yes

Reviewer #2: Yes

3. Has the statistical analysis been performed appropriately and rigorously? 

Reviewer #1: Yes

Reviewer #2: Yes

4. Have the authors made all data underlying the findings in their manuscript fully available?

Reviewer #1: Yes

Reviewer #2: Yes

5. Is the manuscript presented in an intelligible fashion and written in standard English?

Reviewer #1: Yes

Reviewer #2: Yes

6. Review Comments to the Author

Reviewer #1: (No Response)

Reviewer #2: (No Response)

7. PLOS authors have the option to publish the peer review history of their article (what does this mean?). If published, this will include your full peer review and any attached files.

Reviewer #1: **Yes: **Timothy Angelotti MD PhD

Reviewer #2: No

---

## [Editor Report · Acceptance letter]

15 Sep 2023

PONE-D-23-07486R2 

Towards a potent and rapidly reversible Dexmedetomidine-based general anesthetic 

Dear Dr. Xie:

I'm pleased to inform you that your manuscript has been deemed suitable for publication in PLOS ONE. Congratulations! Your manuscript is now with our production department. 

Kind regards, 

on behalf of

Dr. Silvia Fiorelli 

Academic Editor

PLOS ONE